# REEP5 depletion causes sarco-endoplasmic reticulum vacuolization and cardiac functional defects

Shin-Haw Lee [1,2,16], Sina Hadipour-Lakmehsari [1,2,16], Harsha R. Murthy[3,4], Natalie Gibb[3,4], Tetsuaki Miyake[2,14], Allen C.T. Teng[1,2], Jake Cosme [1,2], Jessica C. Yu[1,5], Mark Moon[2,6], SangHyun Lim[7,8], Victoria Wong[7], Peter Liu[6], Filio Billia [9], Rodrigo Fernandez-Gonzalez [1,5], Igor Stagljar[7,8,10,11], Parveen Sharma [2,15], Thomas Kislinger [12,13], Ian C. Scott[3,4] & Anthony O. Gramolini [1,2 ✉]

The sarco-endoplasmic reticulum (SR/ER) plays an important role in the development and progression of many heart diseases. However, many aspects of its structural organization remain largely unknown, particularly in cells with a highly differentiated SR/ER network. Here, we report a cardiac enriched, SR/ER membrane protein, REEP5 that is centrally involved in regulating SR/ER organization and cellular stress responses in cardiac myocytes. In vitro REEP5 depletion in mouse cardiac myocytes results in SR/ER membrane destabilization and luminal vacuolization along with decreased myocyte contractility and disrupted $Ca^{2+}$ cycling. Further, in vivo CRISPR/Cas9-mediated REEP5 loss-of-function zebrafish mutants show sensitized cardiac dysfunction upon short-term verapamil treatment. Additionally, in vivo adeno-associated viral (AAV9)-induced REEP5 depletion in the mouse demonstrates cardiac dysfunction. These results demonstrate the critical role of REEP5 in SR/ER organization and function as well as normal heart function and development.

[1] Translational Biology and Engineering Program, Ted Rogers Centre for Heart Research, Toronto, ON M5G1M1, Canada. [2] Department of Physiology, Faculty of Medicine, University of Toronto, Toronto, ON M5S1M8, Canada. [3] Program in Developmental and Stem Cell Biology, The Hospital for Sick Children, Toronto, ON M5G1X8, Canada. [4] Department of Molecular Genetics, Faculty of Medicine, University of Toronto, Toronto, ON M5S1M8, Canada. [5] Institute of Biomaterials and Biomedical Engineering, University of Toronto, Toronto, ON M5S3G9, Canada. [6] Ottawa Heart Institute, Ottawa, ON K1Y4W7, Canada. [7] Donnelly Centre, University of Toronto, Toronto, ON M5S1M8, Canada. [8] Department of Biochemistry, Faculty of Medicine, University of Toronto Canada, Toronto, Canada. [9] Toronto General Research Institute, University Health Network, Toronto, ON M5G2C4, Canada. [10] Department of Molecular Genetics, Faculty of Medicine, University of Toronto, Toronto, Canada. [11] Mediterranean Institute for Life Sciences, Split, Croatia. [12] Department of Medical Biophysics, University of Toronto, Toronto, Canada. [13] Princess Margaret Cancer Centre, Toronto, ON M5G1L7, Canada. [14] Present address: Department of Biology, Faculty of Science, York University, Toronto, ON M3J1P3, Canada. [15] Present address: Institute of Translational Medicine, University of Liverpool, Liverpool L693BX, UK. [16] These authors contributed equally: Shin-Haw Lee, Sina Hadipour-Lakmehsari. ✉email: anthony.gramolini@utoronto.ca

The sarco-endoplasmic reticulum (SR/ER) is a multifunctional organelle responsible for many essential cellular processes in eukaryotic cells, including protein translation, lipid synthesis[1], $Ca^{2+}$ cycling[2], protein trafficking[3], and organelle–organelle communication[4]. The ER is a lipid-bilayer extension of the outer nuclear membrane consisting of a continuous peripheral ER network of tubules and interspersed sheets[5]. ER tubules are generated mainly by stabilizing membrane curvature, and previous studies have identified a few key evolutionary conserved protein families involved in the formation of these tubular structures[6]. However, many aspects of its structural organization and function remain largely unknown, particularly in highly differentiated muscle cells. Members of the reticulon and Yop1p/DP1/REEP protein families contain a reticulon-homology domain (RHD) which is essential for inducing and stabilizing high membrane curvature in cross-sections of ER tubules[7–9]. In addition to membrane curvature stabilization, ER tubules are also stabilized by forming a characteristic polygonal network through membrane fusion mediated by the atlastin family of dynamin-related GTPases[10,11].

RHD proteins contain two closely-spaced hairpin integral membrane structures (~35 amino acids in length), which have been proposed to form arc-shaped oligomers, thus essential for bending and stabilizing membrane curvature around the tubules[7,12]. Of all RHD proteins, Yop1p from *Saccharomyces cerevisiae* has been the most well-studied. Vertebrate homologs of Yop1p are the family of receptor expression-enhancing proteins (REEPs) and previous studies demonstrate their vital roles in trafficking the odorant receptor[13] and G-protein coupled receptors to the plasma membrane[14]. Despite the association of REEPs' RHD domains to ER network formation, the precise role of REEPs in ER formation, maintenance, and responses to ER stress remains poorly understood. So far, six mammalian REEP homologs have been identified, REEP1 and REEP2 are neuro-enriched in mice[15] and have been linked to hereditary spastic paraplegia in patients and transgenic mice[16,17]. REEP3 and REEP4 are required for mitotic spindle organization in proliferative cells[18]. Mutations in REEP6 have been linked to human retinopathies[19,20]. The role of REEP5, in comparison, remains largely unknown.

Instabilities in ER structure and function lead to ER stress, unfolded protein response, ER-associated degradation, and autophagy[21]. In excitable muscle cells, their ER structures have adapted to handle a large concentration of $Ca^{2+}$, important for regulated release of $Ca^{2+}$ into the cytoplasm for muscle contraction. This specialized smooth ER, termed the SR, evolved to function in striated muscle[22]. However, differences in protein expression and function between the ER and SR have not been fully determined, resulting in poor characterization and understanding of the formation and function of SR in muscle[22]. The SR has been loosely divided into at least two structural and functional domains termed the longitudinal SR and the junctional SR[23]. Furthermore, different regions of the SR have specialized to perform specific functions with respect to the control of the excitation–contraction coupling[24]. It is recognized in patients and animals that longitudinal and junctional SR undergo significant transformation following heart failure[25,26]. While a great deal is known about SR structure and function in terms of cardiac muscle contraction, considerably less is understood about how the SR is formed and maintained.

## Results

### REEP5 is a conserved cardiac-enriched membrane protein.
Our previous proteomic experiments of mouse and human cardiac myocytes, integrated with microarray tissue expression profiles and phenotype ontology information identified poorly characterized, evolutionary conserved, cardiac-enriched membrane proteins[27]. Rank-ordered evaluation of these protein candidates identified that REEP5 was one of these most highly ranked proteins. Accordingly, we investigated the role of REEP5 in the cardiac myocyte.

Given its identification in both mouse and human myocyte proteomic membrane isolations[27], we first performed a detailed multispecies amino-acid sequence analysis of REEP5 which showed 96% homology between human and mouse REEP5 and 73% between human and zebrafish (Fig. 1a). Bioinformatics analysis of extensive phylogeny further demonstrated significant clustering of mammalian REEP5 within the REEP family (Supplementary Fig. 1a).

To verify human REEP5 membrane topology, we used the multi-algorithm prediction tool TOPCONS (http://topcons.cbr.su.se). All six prediction algorithms within this tool determined that the N-terminus was cytosolic for human REEP5, with 2 or 4 predicted transmembrane helices (Fig. 1b). To confirm these predictions for human REEP5, we employed a membrane yeast two-hybrid (MYTH) system, as previously described[27,28] (Fig. 1c); which takes advantage of the ability of ubiquitin fragments (NubG and NubI) reconstitution, N-terminal TF transcription factor tagged (TF-Cub) bait protein and ubiquitin fragments tagged (NubI) prey protein were tested for protein–protein interactions. To determine REEP5 membrane topology at the ER, a noninteracting yeast ER integral membrane protein Ost1p was fused at its C-terminus with NubI and NubG (mutated form) which exposed its C-terminus to the cytosol. N-terminally tagged TF-Cub-REEP5 interacted with Ost1p-NubI but not with Ost1p-NubG, indicating that the N-terminus of REEP5 resides in the cytosol (Fig. 1c). Thus, REEP5 consists of four hydrophobic, hairpin transmembrane domains connected by a hydrophilic segment with both the N- and C-termini facing the cytosol (Fig. 1d). This membrane topology model is entirely consistent with models of Yop1p and other REEP homologs[6,29,30].

To assess cardiac enrichment of REEP5, we used publicly available comprehensive tissue specific human RNASeq datasets (Human Proteome Map) to determine REEP5 expression levels across different tissues[31]. These data demonstrate broad REEP5 expression across several different tissues at both fetal and adult stages, with an apparent increase in expression during development. Importantly, of all the REEP family members, REEP5 was significantly enriched in the fetal and adult heart despite its broad expression across many mouse tissues (Fig. 1e). Immunoblots of diverse tissues in the adult mouse using solubilization buffer containing 6 M urea further demonstrated highest expression in the ventricle (Fig. 1f). Abundant expression was also detected in skeletal muscle (gastrocnemius) and kidney. Apparent REEP5 monomers (17 kDa) and trimers (51 kDa) were predominantly expressed in the ventricle, while REEP5 dimers (34 kDa) were seemingly the predominant form in the atria, suggesting functional difference in REEP5 oligomerization status. Since antibody specificity must always be evaluated carefully, anti-REEP5 antibody specificity was confirmed by blocking antibody with 10 μg of a bacterially expressed, purified 6xHis-tagged REEP5 (Fig. 1g).

We next queried publicly available GEO RNA-seq datasets containing data for human and mouse cardiovascular diseases to determine changes in REEP5 expression. As shown in Fig. 1h, REEP5 levels changed dramatically with several cardiac diseases. A marked decrease in REEP5 mRNA levels was seen in severe dilated cardiomyopathy (DCM) in the mouse and in one human dataset, albeit not significantly. Myocardial infarction (MI) in the mouse and ischemic cardiomyopathy in the human showed a downregulation of REEP5. Similarly, pressure overload-induced

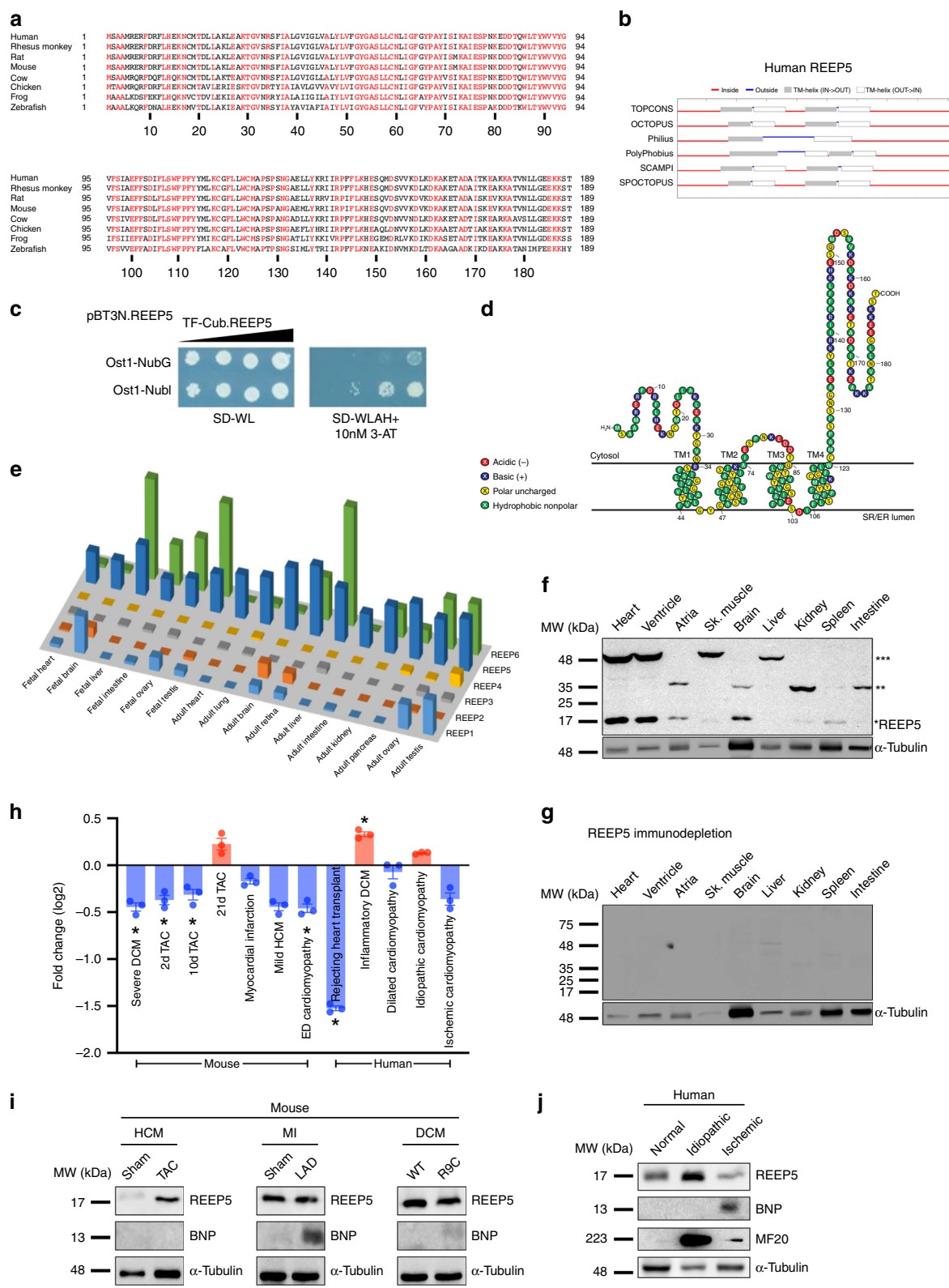

heart failure via transverse aortic constriction (TAC) led to a downregulation of REEP5 at 2 and 10 days post surgery; however, a moderate increase was seen after 21 days. By contrast, inflammatory DCM in humans showed an upregulation of REEP5 RNA levels whereas rejected cardiac allografts in human showed the greatest decrease in REEP5 levels. These results showed that REEP5 levels appear to be dynamic under certain cardiac conditions and suggest a potential role of REEP5 in the progression of heart diseases, although the specific mechanisms and directions of changes are not well defined.

Next, we performed immunoblots using mouse cardiac tissue from a phospholamban (PLN) R9C DCM model[32], 4-week post TAC heart failure model[33], and 2-week post LAD ligation MI model[34]. A marked increase in REEP5 levels was seen in the 4-week TAC mouse hearts whereas REEP5 protein levels were decreased in the LAD ligation MI model (Fig. 1i). Brain

**Fig. 1 REEP5 is an evolutionarily conserved, muscle-enriched membrane protein. a** A multispecies alignment of REEP5 from vertebrates. **b** Prediction of human REEP5 protein topography generated by TOPCONS. **c** Membrane yeast two-hybrid assay of REEP5 membrane topology. SD-WL is yeast media that lacks tryptophan and leucine and selects for cells that contain both bait and prey plasmids. SD-WLAH +10 nM 3-AT is yeast media that lacks tryptophan, leucine, adenine, and histidine and selects for cells in which bait and prey are interacting. **d** Predicted membrane topology model of human REEP5 generated by modification of a T(E)Xtopo output in Protter (http://wlab.ethz.ch/protter). **e** REEP5 mRNA transcript levels obtained from Human Protein Atlas across various mouse tissues. **f** Immunoblot of REEP5 protein expression in mouse tissues. Asterisks to the right indicate the number of predicted REEP5 oligomers detected based on anticipated molecular weight. **g** Immunodepletion of REEP5 antigens with bacterially expressed 6xHis-REEP5. **h** GEO RNA-seq datasets demonstrate changes in REEP5 expression across various mouse and human cardiovascular diseases; * indicates a statistically significant $p < 0.05$ in a Tukey's multiple comparison analysis. Data are presented as mean ± SEM with $n = 3$ biologically independent measurements. **i** Immunoblot analysis of REEP5 and BNP expression from cardiac tissue of hypertrophic cardiomyopathy (HCM), myocardial infarction (MI), and dilated cardiomyopathy (DCM) mouse models. **j** Immunoblot analysis of REEP5, BNP, and MHC expression from human cardiac samples of normal, idiopathic, and ischemic cardiomyopathy. Source data containing original uncropped immunoblots are provided as a Source Data file.

natriuretic peptide (BNP), a robust biomarker for heart failure, dramatically increased in the MI model. PLN R9C mutant DCM mice hearts showed decreased REEP5 levels. BNP levels were very slightly increased in the R9C hearts. Immunoblotting analysis was also carried out in human cardiac samples of idiopathic and ischemic cardiomyopathy. In idiopathic cardiomyopathy, REEP5 and myosin heavy chain β (MF20) levels were elevated. In contrast, REEP5 was downregulated in ischemic cardiomyopathy; in agreement with the GEO data (Fig. 1j).

**REEP5 localizes to the myocyte SR and j-SR membrane.** We next performed IF analysis of endogenous REEP5 in cultured mouse neonatal cardiac myocytes (CMNCs) (Fig. 2a) and acutely dissociated primary adult mouse cardiac myocytes (Fig. 2b, c). As shown in Fig. 2a, REEP5 demonstrated a consistent SR/ER striated staining pattern in CMNCs, with sarcomeric doublet REEP5 patterns observed flanking the z-disks highlighted by F-actin staining, suggesting its localization to the junctional-SR. IF co-staining between REEP5 and SERCA2, a known cardiac SR protein, demonstrated a very high degree of co-localization between the two proteins (Fig. 2b). Further orthogonal and three-dimensional reconstructive analyses confirmed strong co-localization with SERCA2 (Pearson co-localization coefficient of 0.59 ± 0.02; data presented as mean ± SEM) in isolated adult ventricular cardiac myocytes (Fig. 2c), indicating REEP5 localization to the SR in cardiac myocytes. In adult mouse ventricular cardiac myocytes, three-dimensional reconstructive analysis demonstrated a Pearson's co-localization coefficient of 0.64 ± 0.04 between REEP5 and α-actinin, indicating a strong co-localization (Supplementary Fig. 2a, b). Further statistical analysis revealed an average of 99.8 ± 0.1% of α-actinin co-localized with REEP5 while 58.4 ± 1.0% of REEP5 co-localized with α-actinin, suggesting REEP5 localization to the j-SR, but also in patterns indicative of longitudinal SR. Additional analyses revealed strong co-localization between REEP5 and Ryanodine Receptor 2 (RyR2) (Pearson coefficient of 0.76 ± 0.04), and between REEP5 and triadin, with a Pearson coefficient of 0.68 ± 0.12 (Supplementary Fig. 2c, d). These results validate REEP5's localization to the j-SR membrane that is closely tethered to the cell membrane and contractile machinery in adult striated muscles.

**In vitro REEP5 depletion results in SR dysfunction.** To characterize the function of REEP5 in vitro, CMNCs were transduced with a lentiviral REEP5-shRNA construct. Temporal analysis of REEP5 protein levels by immunoblot showed ~60% knockdown was achieved 48 h post transduction with knockdown of 70% following 96 h (Fig. 3a, b). Confocal imaging of transduced cells revealed pronounced vacuolization and disorganization of the SR/ER visualized with ER-Tracker (Fig. 3c). Quantitatively, vacuolization of the SR/ER was observed as soon as 24 h post transduction and observed in nearly all cells at 96 h. To confirm that

the observed vacuolization was not a result simply of dying cells and cell death, we treated cultures with the caspase inhibitor, 100 μM z-vad-fmk, as described previously[35]. Under these conditions, luminal vacuoles were still observed in REEP5 shRNA-infected CMNCs in the presence of z-vad-fmk, ruling out that these morphological changes are simply a result of cell death (Supplementary Fig. 3a), rather they appear upstream of any activated caspase activities.

ER stress plays important roles in contributing to cardiac diseases[36]. To determine if REEP5 depletion-induced structural abnormalities were related to functional impairment, we first assayed for ER stress activity and cell viability. CMNCs were transduced with REEP5-shRNA and intracellular oxidative stress was measured via staining for reactive oxygen species (ROS), reflective of ER stress (Fig. 3d). CellRox fluorescence demonstrated a twofold increase in ROS levels with REEP5 depletion (Fig. 3d). Tunicamycin (5 μg/ml) was used as a cell stress inducer, which increased ROS levels further compared to wild-type CMNCs treated with tunicamycin. Further analysis of ER stress markers, GRp78, GPp94, and ATF4 indicated increased ER stress with REEP5 depletion with/without tunicamycin (Fig. 3f). To assess cell viability, MTT assays were conducted to measure cell metabolic state upon REEP5 depletion and demonstrated a significant decrease in cell viability in cardiac myocytes upon REEP5 depletion (Fig. 3e). Similarly, increased cleaved caspase-12 levels upon REEP5 depletion were observed, indicating activation of ER-dependent apoptosis[37] (Fig. 3f). In addition, we performed mitochondrial membrane potential (MMP) measurements in cardiac myocytes and showed that REEP5 depletion resulted in significant dissipation of the mitochondrial inner membrane electrochemical potential, indicating impaired cellular health and function in the cardiac myocyte (Fig. 3g).

**REEP5 depletion leads to compromised myocyte contractility.** IF of virally mediated REEP5 depletion in adult cardiac myocytes revealed prominent SR vacuoles as shown by REEP5 and RyR2 immunostaining (Fig. 4a). Three-dimensional reconstruction of the SR marked by RyR2 staining further demonstrated the vacuolated SR in REEP5 depleted myocytes as opposed to the striated, interconnected SR network in the control myocytes (Fig. 4b). Transmission electron microscopy (TEM) analysis revealed that REEP5 depleted adult cardiac myocytes showed significant disruption of SR integrity, with deformed SR membranes and apparent vacuoles, along with deformed cardiac t-tubules, compared to control cells (Fig. 4c). Next, we performed optical imaging edge analysis of REEP5 depleted adult cardiac myocytes to measure myocyte contractility of spontaneously contracting cells (Fig. 4d). REEP5 depleted myocytes demonstrated a significant impairment in fractional shortening of the myocytes (0.3 ± 0.1% compared to 12.4 ± 1.2% fractional shortening per contractile event in the control myocytes; data

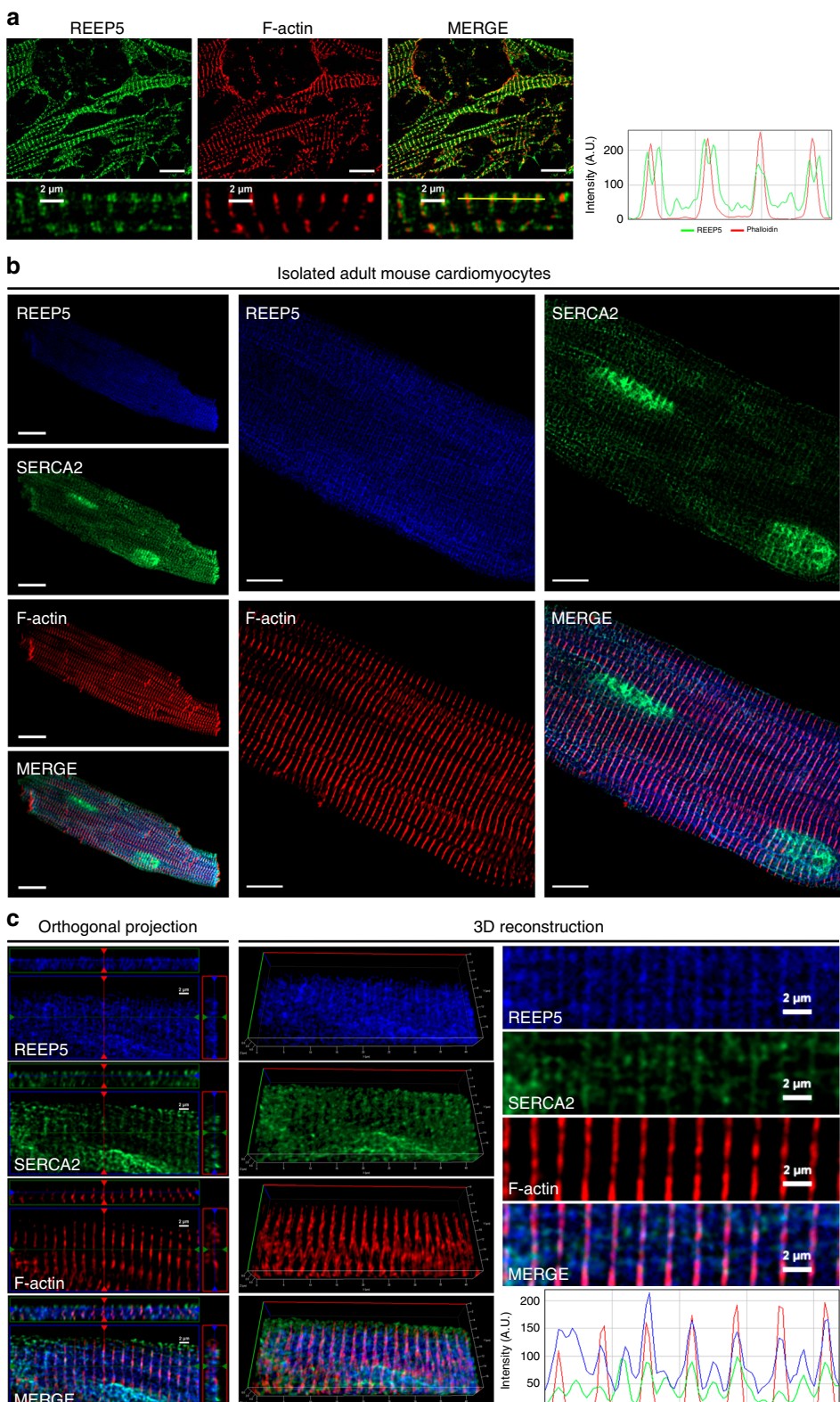

**Fig. 2 REEP5 expression shows consistent SR staining pattern in cardiac myocytes. a** Immunofluorescence analysis of endogenous REEP5 expression (green) and phalloidin-rhodamine staining (red) in CMNCs. Right panel, line-scan analysis (from yellow line) demonstrates SR expression pattern in CMNCs. Scale, 10 μm. **b** Immunofluorescence of isolated adult mouse cardiac myocytes with REEP5 (blue) co-stained with SERCA2 (green) and phalloidin-rhodamine (red). Scale, 20 μm (left panel), 10 μm (right panel). **c** Orthogonal projection, three-dimensional reconstructive and line-scan analyses demonstrates co-localization between REEP5, SERCA2, and phalloidin signal. (Left panel): the top panels represent cell imaging in the x–z plane, while side panels represent cell imaging in the y–z plane. All images shown are representative of approximately 40–50 total images captured per condition, $n = 3$ independent biological replicates.

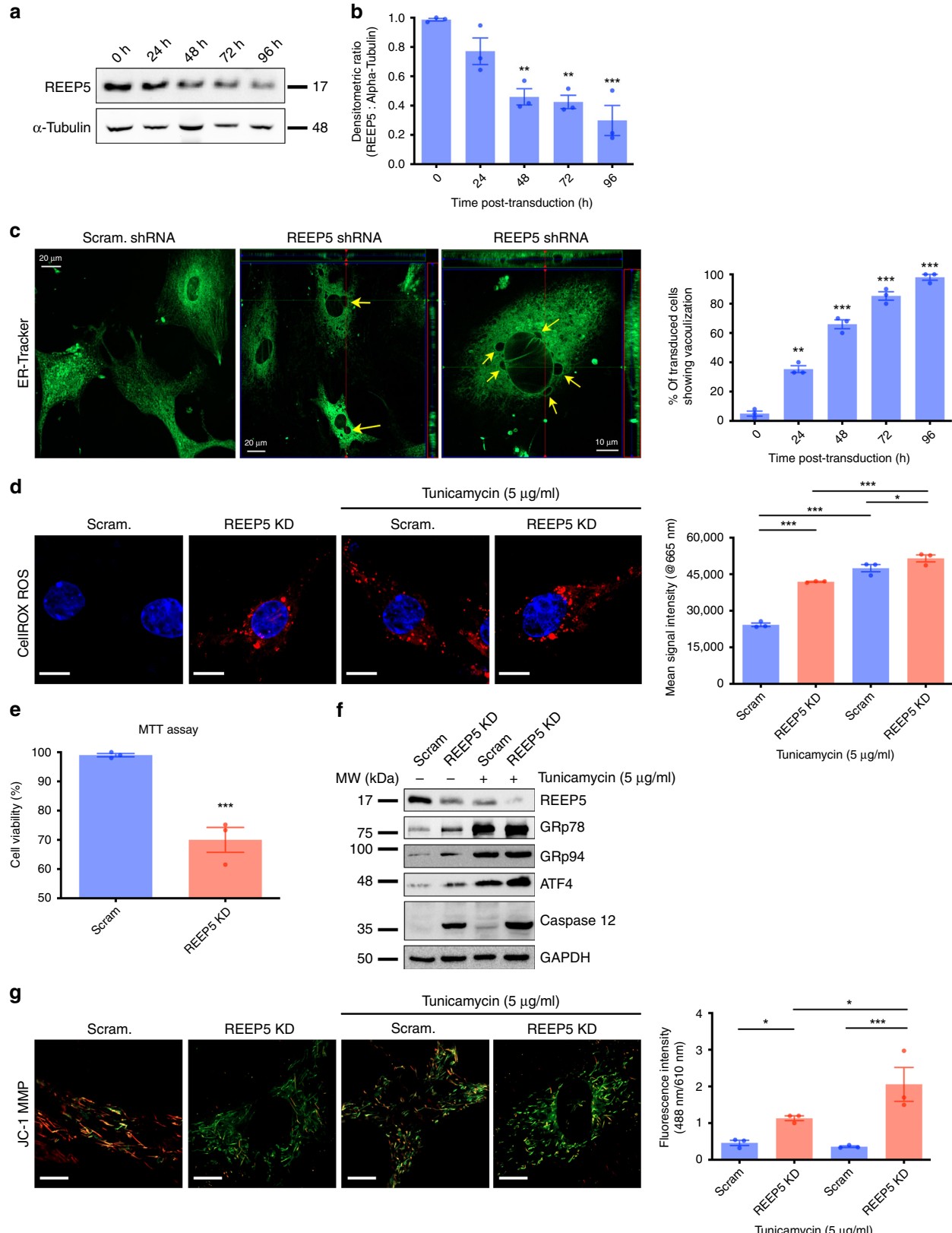

presented as mean ± SEM), suggesting severe functional impairment of myocyte contractility (Fig. 4d, right) with a significant reduction in the recorded events (0.27 ± 0.07 Hz in scram compared to average null values (below detectable threshold) in REEP5 KD; $p = 0.0194$). In addition, $Ca^{2+}$ sensor Fura2-AM measurements were performed to characterize $Ca^{2+}$ cycling in REEP5 depleted myocytes (Fig. 4e). A significant decrease in the number of spontaneous $Ca^{2+}$ transients (1.07 ± 0.05% in scram vs. 0.47 ± 0.03% in REEP5 KD; $p = 0.0006$) and an increased peak amplitude (25 ± 1.5% in scram controls vs. 58.6 ± 1.8% in REEP5 KD; $p = 0.0002$) of these transients were observed in REEP5 depleted myocytes (Fig. 4e, right).

**Fig. 3 In vitro REEP5 depletion in cardiac myocytes results in SR/ER membrane destabilization and dysfunction. a** Immunoblotting analysis of lentiviral-mediated REEP5 depletion in CMNCs at 0, 24, 48, 72, 96 h post transduction. **b** Quantitative analysis of REEP5 expression levels showed 60% reduction 48 h after REEP5 shRNA transduction. Quantification was done under ×40 objective lens and approximately 40–50 cells were scored for each experimental condition, $n = 3$ independent biological replicates; data are presented as mean ± SEM. **c** Confocal imaging of CMNCs stained with ER-tracker showed SR/ER vacuoles (yellow arrows) 48 h post viral transduction. Vacuoles were observed starting 24 h post transduction, peaking after 96 h post lentiviral infection with REEP5 shRNA. Quantification was done under ×40 objective lens and approximately 30–40 cells were scored for each experimental condition, $n = 3$ independent biological replicates; data are presented as mean ± SEM. **d** Confocal imaging of CMNCs stained with CellROX oxidative stress dye 48 h post viral transduction with REEP5 shRNA. Scale, 10 μm. Spectrophotometric analysis showed a marked increase in ROS levels following shRNA-mediated REEP5 depletion in the presence and absence of tunicamycin, $n = 40$–50 cells examined under ×40 objective, $n = 3$ independent biological replicates; data are presented as mean ± SEM. **e** Cardiac myocyte cell viability levels 48 h post viral transduction with REEP5 shRNA measured by MTT assay, $n = 3$ independent biological replicates; data are presented as mean ± SEM. **f** Immunoblotting analysis of REEP5, ER stress markers (GRp78, GRp94, and ATF4), and ER-dependent apoptosis marker (caspase 12) expression levels upon REEP5 depletion in the presence or absence of tunicamycin. **g** Confocal imaging of CMNCs stained with mitochondrial membrane potential dye JC-1 48 h post viral transduction with REEP5 shRNA. Scale, 10 μm, $n = 40$–50 cells examined under ×40 objective lens over 3 independent experiments. Asterisks indicate a statistically significant $p$ value in a Tukey's multiple comparison analysis where *$p < 0.05$, **$p < 0.01$, and ***$p < 0.001$; data are presented as mean ± SEM. Source data containing original uncropped immunoblots are provided as a Source Data file. All images shown are representative of approximately 30–40 total images captured per condition.

**The C-terminal domain of REEP5 contributes to SR integrity.** To study the biochemical properties of REEP5, we created two tagged REEP5 constructs (Fig. 5a). REEP5 was fused to a V5 tag followed by 6xHis tag for affinity purifications using Ni-NTA beads; alternatively, CnVA-tagged[38] (StrepII-6xHis, 3xFlag) REEP5 was generated and used. Immunoblot analysis showed robust expression of both constructs in HEK293 cells (Fig. 5b, d). Interestingly, addition of increasing dithiothreitol (DTT) concentrations led to dose-dependent decrease of observed REEP5 dimers (Fig. 5c), suggesting disulfide bridge(s) may be required for this dimerization. Moreover, overexpression of recombinant proteins CnVA-REEP5 and REEP5-V5-6xHis followed by immunoprecipitation using anti-V5 antibody identified CnVA-REEP5 in the elution, indicating homodimerization of REEP5 molecules (Fig. 5d). Our results are consistent with previous yeast experiments that showed the formation of DP1/Yop1p immobile oligomers[39]. This, however, does not rule out the presence of heterodimers of REEP5 with other proteins, or other REEP isoforms. Furthermore, expression of a fluorescently tagged dimerized REEP5 construct (REEP5-REEP5-EYFP) in $C_2C_{12}$ myoblasts marked the ER network more robustly, suggesting functionality of REEP5 dimers (Supplementary Fig. 4a).

The N- and C-termini cytosolic domains represent the major differences between REEP1-4 and REEP5-6 subfamilies[30]. To determine the importance of various domains in REEP5 to its function, we constructed various EYFP tagged REEP5 truncated mutants (Fig. 5g). Immunoblotting revealed that stable, sodium dodecyl sulfate (SDS)-resistant dimers were detected in $C_2C_{12}$ myoblasts expressed with REEP5-EYFP, REEP5X2-EYFP, and Δ1–36 REEP5-EYFP (Fig. 5e). Interestingly, loss of REEP5 dimers was observed in cells expressing Δ114–189 REEP5-EYFP and Δ1–36/114–189 REEP5-EYFP, suggesting that the cytosolic carboxyl terminal domain is required for the formation of stable dimers. Expression of the carboxyl-terminal truncated mutant of REEP5 (Δ114–189 REEP5-EYFP) in $C_2C_{12}$ myoblasts led to a vacuolated ER network as observed with REEP5 depletion in cardiac myocytes (Figs. 3c and 4a). To confirm the origins of the vacuoles, recombinant mCherry fusion proteins with mitochondria targeting peptide (mCherry-mito) or with a ER lumen retention peptide (mCherry-KDEL) were co-expressed with Δ114–189 REEP5-EYFP in $C_2C_{12}$ myoblasts (Fig. 5h). Co-expression of mCherry-KDEL, as an ER–luminal protein, occupied the entire vacuoles generated by expression of Δ114–189 REEP5-EYFP, indicating that the observed vacuoles originate from the ER membrane. Consistent with this hypothesis, expression of Δ1–36 REEP5-EYFP did not affect the morphology of the ER and co-localized strongly with mCherry-

KDEL, Δ114–189 REEP5-EYFP and Δ1–36/114–189 REEP5-EYFP significantly disrupted the peripheral ER morphology and caused prominent luminal swelling and vacuolization (Fig. 5i). To test whether disruption of the ER network affects microtubule dynamics, we co-expressed Δ114–189 REEP5-EYFP and mCherry-α-Tubulin in $C_2C_{12}$ myoblasts and showed intact microtubule organization in Δ114–189 REEP5-EYFP transfected cells, suggesting the SR/ER shaping role of REEP5 is independent of microtubule polymerization (Supplementary Fig. 4b).

**REEP5 interacts with known SR/ER shaping proteins.** Mass spectrometry-based studies identified REEP5-associated proteins as nickel-His purification of CnVA-REEP5 in HEK293 (Fig. 6a) resulted in the identification of several members of the reticulon (RTN) and atlastin (ATL) families of proteins with a marked increased fold-change enrichment in purified REEP5-associated membrane complexes (Fig. 6b). Assessment of ACTC1, MYL6, and GAPDH as negative controls in both conditions demonstrated similar average peptide intensity with minimal fold-change difference detected. Notably, we identified RTN4 (Nogo-A/B), RTN3, and ATL2 solely in the REEP5 transfected cells, while ATL3 and CKAP4, were detected under non-transfected and transfected conditions, although at a 15-fold and 5-fold enrichment in the REEP5 transfected cells were detected, respectively.

To identify which RTN, ATL, and CKAP isoforms are highly expressed in cardiac myocytes, we again analyzed publicly available transcriptomic datasets from the Human Proteome Map[31] for RTN, ATL, and CKAP families of proteins. Of particular interest, Nogo-A/B/RTN4, ATL3, and CKAP4/Climp-63 were significantly enriched in the fetal and adult heart, similar to REEP5 (Fig. 6c).

Previous experiments demonstrated the interaction between RTN1 and DP1/Yop1p in yeast and showed subsequent oligomer formation was essential for tubular ER localization and ER tubule formation[39]. Moreover, immunoblots of the REEP5 nickel-His purification eluate confirmed the presence of RTN4, ATL3, and CKAP4 (Fig. 6d). Here, we overexpressed CnVA-REEP5 and Myc-RTN4 (Nogo-B) plasmids in HEK cells (Fig. 6e), followed by co-immunoprecipitation with anti-REEP5 (Fig. 6f, left) or anti-Nogo-A/B antibodies (Fig. 6f, right). In both experiments, Nogo isoforms were coprecipitated with REEP5 confirming the physical interaction between these two proteins, at least in an overexpression system (Fig. 6f). Next, co-immunoprecipitations using mouse ventricular membrane fractions (microsome) were performed. Immunoprecipitating

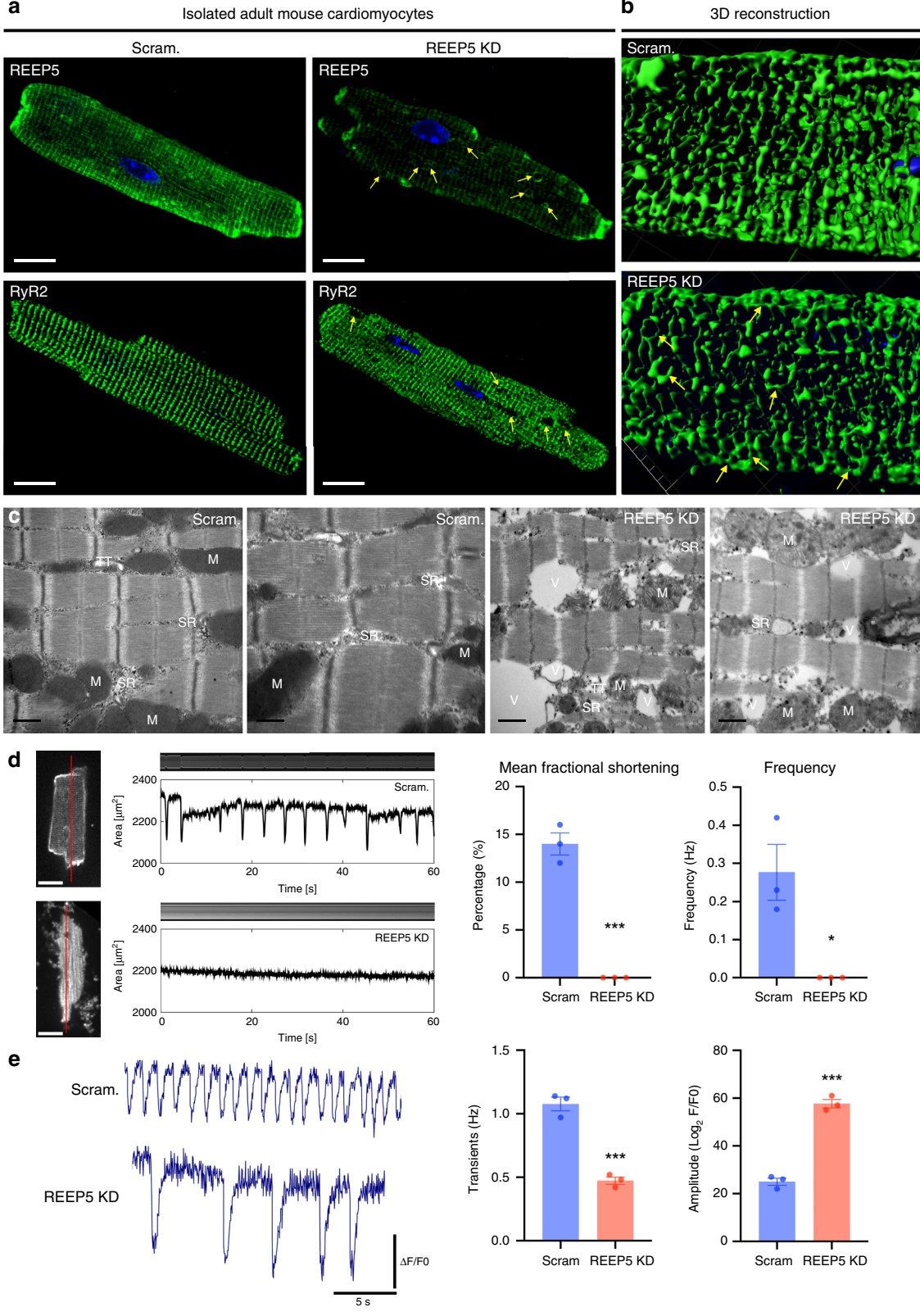

REEP5, followed by immunoblots, showed REEP5 co-precipitated Nogo-A, Nogo-B, ATL3, an ATL3 apparent dimer, and CKAP4 in these lysates (Fig. 6g). REEP5 in a dimer, and oligomer state were detected (Fig. 6g, lower panel). Co-immunoprecipitations in a reverse strategy were also performed (Fig. 6h–j). In these studies, we showed that all four antibodies were able to coprecipitate REEP5, mostly as higher molecular weight dimer and oligomer, along with their respective antigens. Finally, as parallel experiments, co-immunoprecipitation assays performed using total neonatal ventricular cardiac lysate also revealed similar REEP5 interactions with Nogo-A/B, ATL3, and CKAP4 (Supplementary Fig. 5a).

**Fig. 4 In vitro REEP5 depletion leads to SR vacuolization and sarcomeric dysfunction in adult mouse cardiac myocytes. a** Immunofluorescence of adult mouse cardiac myocytes stained with REEP5 and RyR2 48 h post viral transduction with REEP5 shRNA. Scale, 20 μm. **b** Three-dimensional reconstruction analysis of RyR2 staining of the SR in Scram and REEP5-depleted myocytes. **c** Transmission electron microscopy in REEP5-depleted adult mouse cardiac myocytes revealed SR vacuoles and disrupted SR membranes compared to scram controls. M mitochondria, SR sarcoplasmic reticulum, TT T-tubule, V vacuoles. Scale, 0.5 nm. **d** Optical measurements of spontaneous myocyte contractility in scram and REEP5-depleted adult mouse cardiac myocytes revealed a significant decrease in both fractional shortening measurements and frequency in REEP5-depleted myocytes. Left: still images of representative cardiac myocytes. Scale, 20 μm. Red line indicates region of image used to generate kymographs shown above contractile pulses tracings. $n = 30$ cells examined over three independent experiments; data are presented as mean ± SEM. **e** $Ca^{2+}$ imaging of myocytes showing the frequency of $Ca^{2+}$ waves and $Ca^{2+}$ transients amplitude 48 h post viral transduction with REEP5 shRNA, $n = 40$–50 cells examined per condition over 3 independent experiments. Asterisks indicate a statistically significant $p$ value in a Tukey's multiple comparison analysis where *$p < 0.05$ and ***$p < 0.001$; data are presented as mean ± SEM. All images shown are representative of approximately 30–40 total images captured per condition.

**In vivo phenotype analysis of REEP5 depletion in zebrafish.** We next used the zebrafish embryonic model, with its rapid ex utero development, to investigate the role of REEP5 in vivo. We performed IF analysis at 48 and 96 h post fertilization (hpf) to examine the localization of REEP5 expression within cardiac myocytes. Co-staining with zn-8 (localized to the membrane) and DAPI revealed REEP5 expression within both ventricular and atrial cardiac myocytes, with minimal nuclear or membrane localization evident (Fig. 7a).

To determine the function of REEP5 during zebrafish heart development, embryos were injected with a titrated minimum functional dosage of 1 ng of REEP5 morpholino (MO) at the one-cell stage to block REEP5 translation. Reduced REEP5 expression was observed in MO embryos (Supplementary Fig. 6a). Using *Tg (myl7:EGFP)* embryos, demarcating differentiated cardiac myocytes, REEP5 morphant embryos demonstrated abnormalities in heart morphology with a lack of looping at 48 hpf (Supplementary Fig. 6b, c). To determine the specificity of the REEP5 MO sequence, REEP6 (92% homology to REEP5) MO sequence was used as a control. Investigations into REEP6 MO embryos revealed no observable cardiac phenotype, supporting the specificity of REEP5 MO associated phenotypes (Supplementary Fig. 6d). Optical imaging and IF of the REEP5 morphants showed defects in cardiac looping where a linear-oriented 2-chambered heart morphology in REEP5, compared to a fully looped heart in the sham controls (Supplementary Fig. 6e–h). These morphological defects were accompanied by irregularities in heart rhythm suggestive of an atrioventricular heart block. Further optical imaging analysis of cardiac conduction patterns demonstrated arrhythmogenic beating rhythms with asynchronous and skipped heart beats in REEP5 MO embryos compared to controls (Supplementary Fig. 6i and Supplementary Movies 1–3). The regular heartbeat of a control embryo by 96 hpf is a 1:1 ratio of atrial:ventricular contractions. Upon loss of REEP5 function, this ratio became offset with the atrium beating 3× to every ventricular contraction. TEM analysis of control embryos revealed cardiac myocytes with organized sarcomeres displaying well-ordered actin–myosin filaments and intact SR membranes (Supplementary Fig. 6j, k). However, TEM analysis of REEP5 morphants revealed apparent SR membrane vacuolization in zebrafish ventricular tissue along with structural discontinuity between sarcomeres (Supplementary Fig. 6l, m).

In vivo functional rescue experiments by injecting wild-type REEP5 mRNA in REEP5 MO zebrafish embryos showed co-injection efficiently rescued the cardiac dysfunction phenotypes observed in REEP5 MO embryos, resulting in significantly improved heart rate (140 ± 11 bpm in control vs. 56 ± 4 bpm in REEP5 MO vs. 134 ± 4 bpm in REEP5 MO + REEP5 mRNA injected embryos; $p = 0.004$. Data presented as mean ± SEM and heart morphology in the rescued morphants (Supplementary Fig. 6n–p). Further analysis showed reduced AV dys-synchrony phenotype with an apparent healthy ratio of 1:1 atrial:ventricular

contractions in the rescued embryos (0.3 ± 0.3% in control vs. 83 ± 4% in REEP5 MO vs. 10 ± 2% in REEP5 MO + REEP5 mRNA injected embryos; $p < 0.0001$). Immunoblots of REEP5-interacting proteins identified in Fig. 6 were also performed and showed a marked increase in the protein expression levels of ATL3 (0.48 ± 0.23 in control vs. 0.89 ± 0.19 in REEP5 MO embryos; densitometric ratio normalized to GAPDH, $p = 0.04$) (Supplementary Fig. 6q).

To address potential off-target effects of the REEP5 MO mutant and to validate the REEP5 MO model, we used CRISPR/Cas9 to generate a REEP5 loss-of-function mutant in zebrafish[40]. High resolution melt (HRM) analysis of REEP5-targeted gRNA and Cas9 mRNA injected zebrafish embryos ($F_0$) showed a marked temperature shift in the melt curves comparing controls and REEP5 gRNA injected embryos, confirming the cutting efficiency of REEP5-targeted gRNA (Supplementary Fig. 7a). Depletion of REEP5 protein expression was confirmed by subsequent immunoblotting and IF analyses of mutant hearts (Supplementary Fig. 7b, c). To determine the functional and cutting specificity of our designed gRNA, we performed dose response gRNA injection conditions (1×, 2×, 3×) (Fig. 7b–e). We determined 2× gRNA provided high phenotypic penetrance and low toxicity and these injected embryos were used for all subsequent functional analysis. Optical imaging of cardiac morphology and contraction in REEP5 gRNA injected fish showed cardiac developmental defects, displayed as a lack of cardiac looping, as well as atrio-ventricular conduction defects with dysynchronous beating rhythms (Fig. 7f–h). Further quantification showed that more than 40% of the 2× and 3× gRNA injected $F_0$ embryos showed cardiac developmental defects with a linearized heart (Fig. 7i). In all experimental conditions there was reduced heart rate and increased atrio-ventricular dys-synchrony associated with REEP5 gRNA injection. Increasing gRNA concentration led to a greater number of embryos displaying more severe developmental defects with reduced heart rate and impaired trunk circulation (Fig. 7j, k and Supplementary Movies 4–8). These defects were associated with poor survival since 60% of all analyzed REEP5 morphant and crispant mutants embryos did not survive past 7 days, highlighting the importance of REEP5 during embryonic heart development.

Next, we generated germline *reep5* CRISPR homozygous maternal zygotic (MZ) zebrafish mutants. Our sequencing analysis showed successful editing of the *reep5* gene at the gRNA target site, resulting in the generation of *reep5* homozygous CRISPR mutants (Fig. 8a). However, analysis of F4 *reep5* germline CRISPR MZ mutant embryos showed minimal morphological abnormalities nor developmental defects and did not present with the complete lethality we had observed previously (Fig. 8b–e and Supplementary Movie 9).

Short-term treatment with verapamil, a voltage-gated $Ca^{2+}$ channel blocker, has been proposed as a rapid heart failure

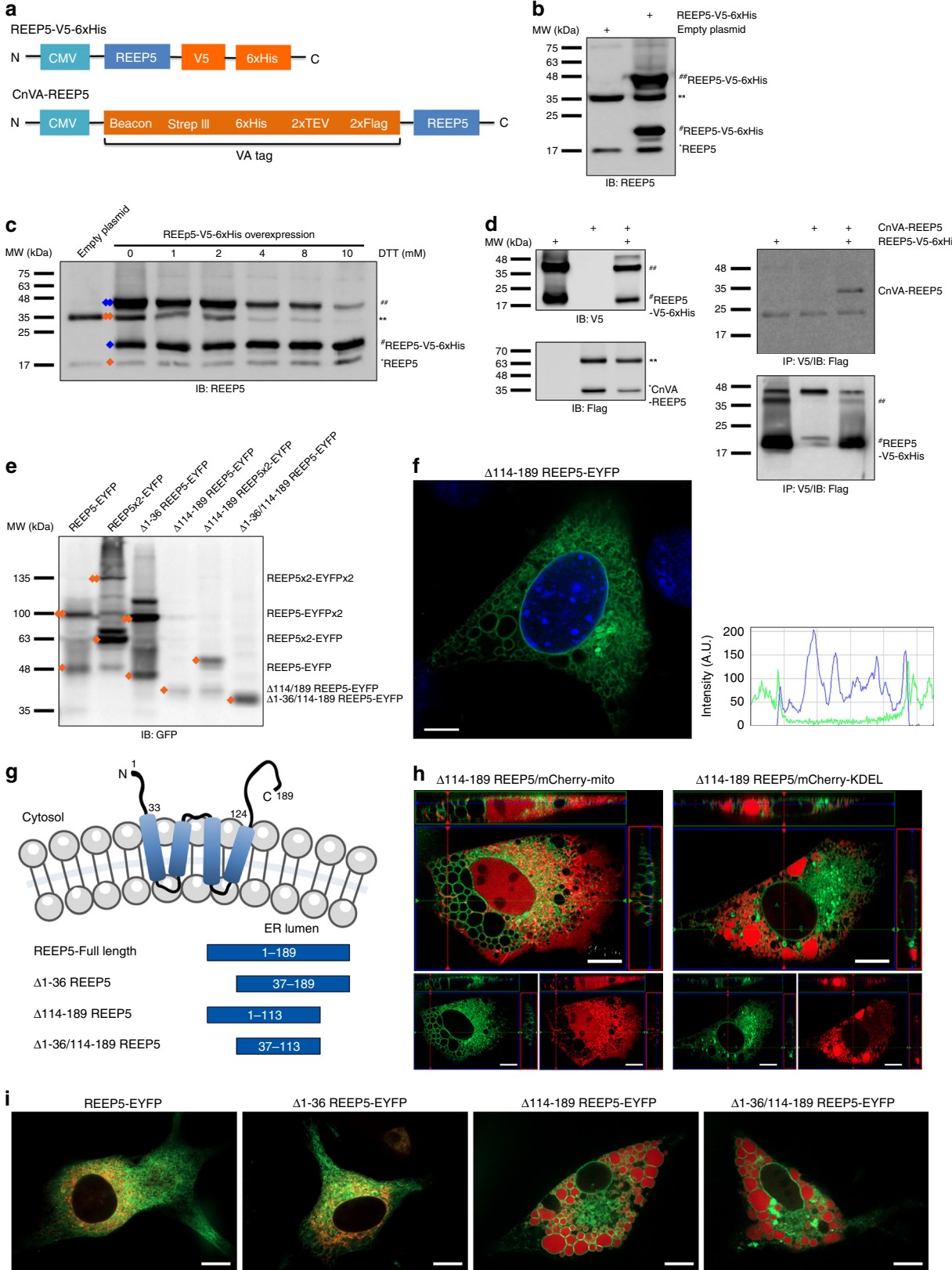

model[41]. Zebrafish at 48 hpf were treated with 0.6 mg/mL verapamil for 30 min to induce a rapid decrease in heart rate and contractility, ultimately leading to heart failure[41]. The *reep5* germline mutants treated with verapamil displayed sensitized cardiac dysfunctional phenotypes highlighted by a marked reduction in heart rate compared to control embryos

($133 \pm 5$ bpm in control vs. $125 \pm 4$ bpm in *reep5* CRISPR fish under resting conditions, compared to $131 \pm 4$ bpm in control vs. $99 \pm 4$ bpm in the *reep5* CRISPR fish with verapamil treatment; $p = 0.003$. Data presented as mean $\pm$ SEM (Fig. 8f). Compensatory network induced by deleterious mutations in CRIPSR germline mutants, but not in zebrafish knockdown models such

**Fig. 5 The C-terminal cytosolic domain of REEP5 is required for stabilizing SR/ER morphology. a** Schematic diagrams of REEP5 constructs used for REEP5 dimerization study shown in Fig. 5b–d. **b** Immunoblot analysis of endogenous REEP5 monomer (~17 kDa) and REEP5 dimer (~34 kDa), and exogenous REEP5 monomer (~22 kDa) and exogenous REEP5 dimer (~44 kDa) in HEK293 transfected with REEP5-V5-6xHis construct. **c** Immunoblots of REEP5 dimer dissociation in response to increasing DTT concentration. Diamonds to the left indicate the detection of endogenous (orange) and exogenous (blue) REEP5 monomers and dimers. **d** Co-immunoprecipitation assays in HEK293 cells. REEP5-V5 and CnVA-REEP5 were transfected into cells and precipitated with V5 or flag antibody. Left, control immunoblot experiments. Right panel, CnVA-REEP5 was immunoprecipitated via anti-V5 antibody in REEP5-V5-6xHis transfected HEK293 cells. **e** Immunoblot analysis of $C_2C_{12}$ cells transfected with REEP5 truncation mutants shows depletion of REEP5 dimers in the absence of the carboxyl terminal domain. **f** Confocal imaging analysis of EYFP (green) and DAPI (blue) staining shows truncation of the carboxyl terminal domain of REEP5 (Δ114–189) causes ER luminal vacuolization in transfected $C_2C_{12}$ myoblasts. Scale, 10 μm. Right, image intensity analysis along axis. **g** Schematic diagram of truncated REEP5 mutant constructs fused with GFP. Amino acid sequences are shown. **h** Live-cell confocal imaging of $C_2C_{12}$ myoblasts expressing the carboxyl-terminus truncated mutant of REEP5 (Δ114–189) and recombinant mCherry-fused mitochondrial targeting signal (mCherry-mito) or luminal ER marker (mCherry-KDEL). Scale, 10 μm. **i** Live-cell confocal imaging of $C_2C_{12}$ myoblasts expressing four truncated REEP5 mutants co-expressed with mCherry-KDEL suggests the importance of the C-terminal cytosolic domain of REEP5 in stabilizing ER membrane curvatures. Scale, 10 μm. Source data containing original uncropped immunoblots are provided as a Source Data file. All images shown are representative of approximately 40–50 total images captured per condition, $n = 3$ independent biological replicates.

as morphants or crispants, has been described previously[42]. Thus, we next examined the possibility of such genetic compensation by injecting REEP5 MO into *reep5* germline mutants and showed that *reep5* mutants injected with REEP5 MO were phenotypically normal compared to overt cardiac abnormalities in the REEP5 morphants (Fig. 8g–j and Supplementary Movie 10). Further quantification showed *reep5* CRISPR mutants were less sensitive to MO injection, with ~2 ± 1% of *reep5* CRISPR mutants vs 65 ± 4% of REEP5 MO embryos showed cardiac developmental defects with a linearized heart (Fig. 8k). Lastly, immunoblots of the REEP5 interacting proteins showed a marked increase in the protein expression levels of RTN4 (0.89 ± 0.16 in control vs. 1.99 ± 0.18 in *reep5* CRISPR fish; densitometric ratio normalized to α-actin, $p = 0.01$), suggesting potential compensatory mechanism regulated by RTN4 upregulation in response to the loss of REEP5 in zebrafish (Fig. 8l).

**In vivo AAV9-mediated REEP5 depletion in mice**. In parallel to REEP5 depletion in zebrafish embryos, we performed intraperitoneal injections of AAV9 virus carrying scrambled shRNA or REEP5 shRNA in neonatal mice (P10). Immunoblotting analysis of REEP5 4 weeks post viral infection showed near depleted levels of REEP5 in the AAV9 REEP5 shRNA injected mouse hearts (Fig. 9a). All AAV9 REEP5 shRNA injected mice developed lethal cardiac dysfunction at 4 weeks post viral transduction and we never observed animals surviving past 5 weeks post injection, compared to healthy AAV9 scram injected littermates.

Hematoxylin and eosin (H&E) and Masson's trichrome staining of the REEP5 depleted myocardium showed general disorganization of the myocardium compared to scram controls (Fig. 9b). Masson's trichrome staining revealed greater fibrotic regions, with significant collagen deposition in the REEP5 depleted ventricular myocardium (Fig. 9b, bottom panels). TEM analysis (Fig. 9c) of control ventricular tissue showed organized sarcomeres displaying well-ordered actin–myosin filaments, whereas REEP5 depleted ventricular myocardium showed disordered sarcomere organization and SR membrane vacuolization, consistent with the observations in REEP5 depleted adult cardiac myocytes and zebrafish embryos (Fig. 9c). Echocardiographic assessment of cardiac function of AAV9 scram shRNA and REEP5 shRNA injected mice was performed (Fig. 9d) and echocardiographic B-mode and M-mode measurements revealed significant myocardial dilatation and cardiac contractile dysfunction (Fig. 9e), with a marked decrease in cardiac ejection fraction 61.0 ± 0.9% in scram vs. 23.1 ± 2.2% in REEP5 KD; $p < 0.0001$, stroke volume (46 ± 2 μL in scram compared to 18 ± 2 μL in REEP5 KD; $p < 0.0001$), and cardiac output (19.0 ± 0.5 mL/min in scram vs. 10.1 ± 1.2 mL/min in REEP5 KD; $p = 0.0002$) in AAV9

REEP5 shRNA injected mice compared to its AAV9 scram injected littermates (Fig. 9e).

REEP5-depleted myocardium had elevated expression levels of ER stress markers GRp78 (1.5 ± 0.2 in scram vs. 2.5 ± 0.5 in REEP5 KD; densitometric ratio normalized to α-actin, $p = 0.13$. Data presented as mean ± SEM, GRp94 (0.05 ± 0.01 in scram vs. 1.06 ± 0.84 in REEP5 KD; $p = 0.27$), XBP1 (0.07 ± 0.01 in scram vs. 0.13 ± 0.02 in REEP5 KD; $p < 0.05$), ATF4 (0.04 ± 0.01 in scram vs 0.79 ± 0.11 in REEP5 KD; $p = 0.002$), and CHOP (0.02 ± 0.01 in scram vs. 0.17 ± 0.05 in REEP5 KD; $p = 0.001$) (Fig. 9f). In line with our previous in vitro studies in cardiac myocytes (Fig. 3f), a marked increase in the expression levels of cleaved caspase 12 (0.03 ± 0.01 in scram vs. 0.25 ± 0.06 in REEP5 KD; $p = 0.004$) was observed, indicating activated ER stress-induced apoptotic activity in the REEP5 depleted ventricular myocardium (Fig. 9f).

Lastly, we measured expression levels of the newly identified REEP5 interacting proteins. Immunoblots revealed significantly increased levels of RTN4 (1.1 ± 0.2 in scram vs. 2.3 ± 0.5 in REEP5 KD; densitometric ratio normalized to α-actin, $p = 0.01$), ATL3 (0.14 ± 0.04 in scram vs. 0.45 ± 0.07 in REEP5 KD; $p = 0.02$), and CKAP4 (0.59 ± 0.04 in scram vs. 1.52 ± 0.31 in REEP5 KD; $p = 0.04$) in REEP5 depleted hearts (Fig. 9g). Interestingly, we also investigated protein levels of these SR/ER shaping proteins in the context of cardiac pressure overload-induced hypertrophy and heart failure and were able to demonstrate consistent and significant elevations in protein levels of REEP5, RTN4, ATL3, and CKAP4 in failing mouse hearts following aortic constriction (see Supplementary Fig. 8a).

## Discussion

This study details a fundamental role of REEP5 in the cardiac myocyte, demonstrates that this protein is essential for SR/ER organization and proper function, identify its protein interactions, and shows that its deletion can result in cardiac functional and developmental defects. REEP5 expression was demonstrated to be particularly muscle-specific with the highest protein expression in the mouse ventricles and skeletal muscle. Our results show that REEP5 expression is detected across the SR network in cardiac myocytes, co-localizing with known SR markers (SERCA2, α-actinin, RyR2, and triadin), thereby regulating SR function in cardiac muscle.

Previous experiments in yeast have established the interactions between REEP1, RTN1, and ATL1 coordinate the generation and maintenance of the tubular ER network[30]. In both mouse neonatal cardiac myocytes and adult cardiac microsomes, we demonstrated physical interactions between REEP5 and other SR/ER structure proteins including RTN4/Nogo-A/B, ATL3, and

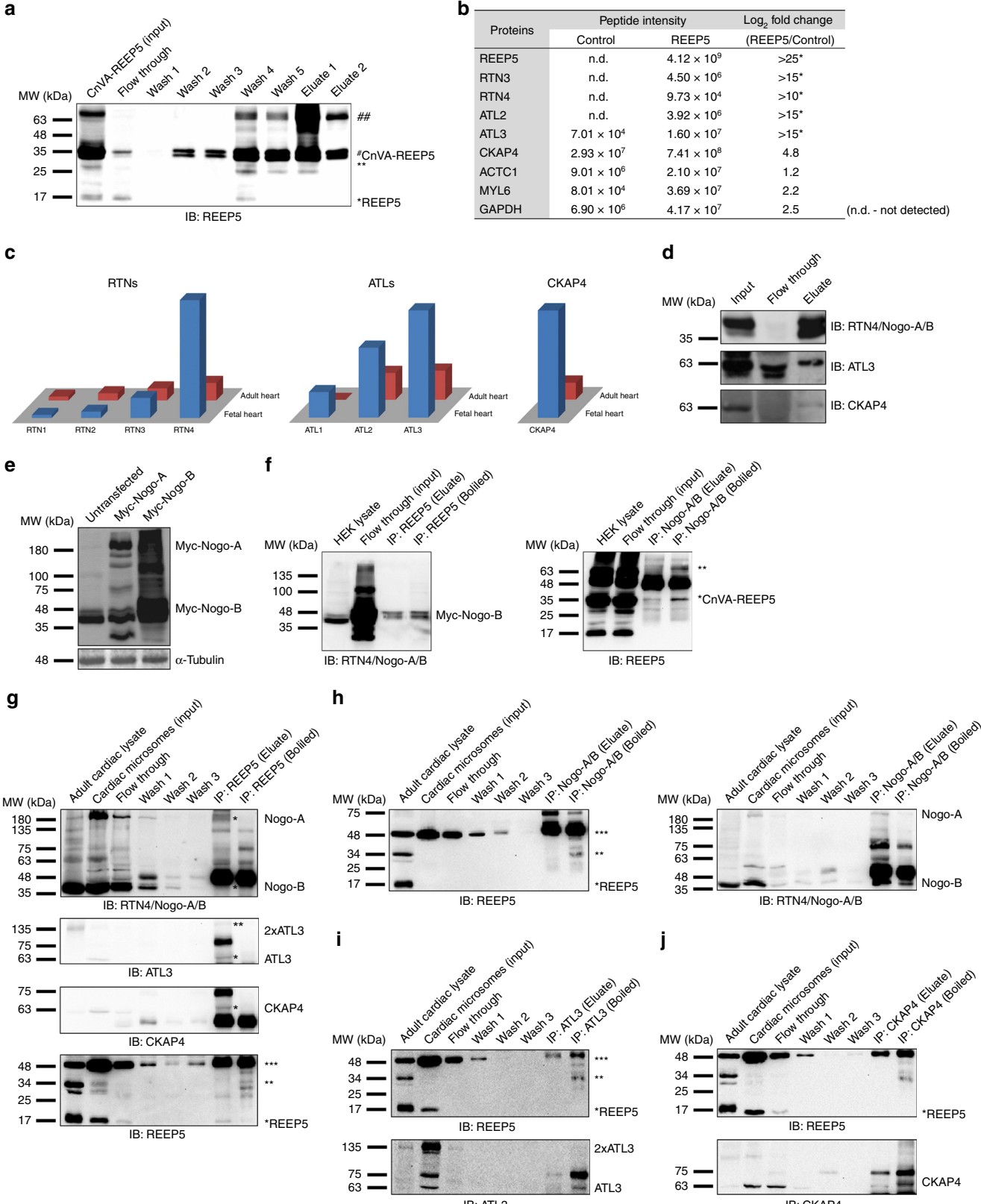

CKAP4. A fine balance between SR/ER structure proteins has been proposed to be critical for generating and maintaining a dynamic SR/ER network as overexpression of membrane curvature stabilizing proteins generates long, unbranched tubules with a marked decrease in tubular three-way junctions and absence of membrane fusion were observed with ATL inactivation[43–45].

Wang et al.[46] further demonstrated the roles and cooperation between ATLs and RTNs in ER three-way junction morphology and maintenance in *Xenopus laevis* eggs. Our results support the interactions of these SR/ER structure proteins in primary cardiac myocytes. Moreover, we show that disruption of these interactions through REEP5 depletion resulted in a marked increase in

**Fig. 6 Mass spectrometry analysis identifies REEP5 interactions with known cardiac SR shaping proteins. a** Immunoblot analysis of nickel-His purification of CnVA-REEP5 from transfected HEK293 cells shows stable expression of CnVA-REEP5 and successful immunoprecipitation of CnVA-REEP5 monomer (~34 kDa) and dimer (~68 kDa). **b** Identification of known SR/ER-shaping proteins as REEP5 interacting proteins by mass spectrometry analysis. Average precursor MS1 peak areas (peptide m/z signal) as defined by iBAQ (intensity based absolute quantification) are shown, $n = 3$ independent mass spectrometry runs. To calculate fold change, average null values (n.d.—not detected) were inputted with a value of 10. RTN reticulon, ATL atlastin, CKAP4 cytoskeleton-associated protein 4, ACTC1 alpha cardiac muscle actin 1, MYL6 myosin light chain 6, GAPDH glyceraldehyde-3-phosphate dehydrogenase. Asterisks indicate a statistically significant $p$ value in a Tukey's multiple comparison analysis where *$p < 0.05$. **c** RNASeq analysis of the RTN, ATL families of proteins, and CKAP4 in human fetal heart and adult heart tissue data using data from Human Protein Atlas. **d** Immunoblot analysis of nickel-His REEP5 immunoprecipitation lysates for RTN4/Nogo-A/B, ATL3, and CKAP4, $n = 3$. **e** HEK293 cells were transfected with myc-tagged Nogo-A and Nogo-B plasmids and detected with myc and α-tubulin antibodies. **f** Co-immunoprecipitation assay of cotransfected HEK293 cells with anti-REEP5 antibody (left panel) and anti-RTN4/Nogo-A/B antibody (right panel) demonstrated interaction between REEP5 and RTN4, $n = 3$. Eluates were collected on ice and loaded directly for blotting, or samples were boiled prior to blotting. **g–j** Co-immunoprecipitation and reverse order co-immunoprecipitation assays with **g** anti-REEP5, **h** anti-RTN4/Nogo-A/B, **i** anti-ATL3, and **j** anti-CKAP4 antibodies in adult mouse cardiac microsomes (input), followed by immunoblots analysis, $n = 5$. Source data containing original uncropped immunoblots are provided as a Source Data file.

the expression levels of RTN4, ATL3, and CKAP4 in mouse ventricular myocardium. Interestingly, a marked increase in the expression of RTN4 was observed in our germline *reep5* CRISPR mutant whereas a clear reduction in the expression levels of ATL3 and CKAP4 were detected. Altogether, these data suggest that REEP5 may act upstream of other ER structure proteins or may possess a more critical role in the heart. While increased RTN4 protein expression may be sufficient to compensate for the loss of REEP5 in CRISPR-based *reep5* null zebrafish embryos, this does not appear to be the case in our in vivo studies in the mouse with the rapid knockdown of REEP5 via viral deletion. Furthermore, REEP5 interaction with a previously unassociated sheet-ER stabilizing protein, CKAP4, may explain the importance of REEP5 in stabilizing membrane curvature at the edges of ER sheets[47]. This interaction may be important for cardiac SR structure as primary cardiac myocytes are more complex than yeast or *X. laevis*.

Mutations and altered protein modifications in many ER morphology proteins including ATL1, REEP1, and REEP2 account for the most common forms of a human neurodegenerative disorder known as the hereditary spastic paraplegias[30,48,49]. In addition, recent studies demonstrate that genetic mutations in REEP6 are responsible for causing retinitis pigmentosa[19,20], an inherited retinal dystrophy characterized by loss of photoreceptors in the retina. REEP5 protein levels in the context of cardiac disease showed increased REEP5 expression in mouse hypertrophic cardiomyopathy and human idiopathic heart disease, indicating a potential role of REEP5 in the progression and development of heart disease. These results suggest that proper REEP5 protein expression contributes to SR/ER integrity and maintenance in cardiac myocytes and may be responsible for altered SR/ER functions and morphological defects in cardiac disease. In fact, in vitro REEP5 depletion in isolated cardiac myocytes resulted in activation of ER stress and the initiation of ER-dependent cell death as seen by caspase-12 activation.

We observed phenotypic discrepancies between our morphant, crispant, and germline CRISPR *reep5* mutant models in zebrafish that we believe can be attributed to the fundamental difference between knockdown and knockout experimental approaches. Morpholino and crispant-driven knockdown experiments in zebrafish embryos allow investigations of acute embryonic phenotypes in embryos whereas the generation of genetically mutated homozygous mutants provides a window for physiological adaptation or genetic compensation. Previously, activation of genetic compensation only in response to deleterious mutations, but not in knockdown-driven morphants or crispants, has been observed in zebrafish[42]. Similar compensatory phenomena, as well as differences between knockdown and knockout models, have also been reported in yeast[50]. In fact, a significant

upregulation of RTN4 in the *reep5* germline mutant but not in REEP5 morphants suggests that the consequence of germline *reep5* deletion is perhaps masked by the upregulation of RTN4 in the germline *reep5* mutant hearts.

Nevertheless, our REEP5 crispant data in the zebrafish embryos are consistent with in vitro REEP5 depletion experiments in the myocyte and in vivo depletion in the mouse. In both cases, altered cell morphology and abnormal heart morphology were observed, highlighting the fundamental role of REEP5 in cardiac biology. Of course, abnormal fish phenotype must always be approached in a very critical manner[51]. However, given that we observed consistent phenotypes within the myocytes in vitro, and zebrafish and mouse hearts in vivo, it is highly probable that the observed phenotypes are a direct result of REEP5 deficiency. Our results are supported by a recent study that generated CRISPR/Cas9-mediated inactivation of REEP5 in rat ventricular myocardium with a preliminary characterization that showed depressed cardiac contractility in pressure–volume loops[52]. Interestingly, inactivation of REEP5 in rats displayed a weaker and nonlethal cardiac phenotype compared to our REEP5-depleted mouse model. This discrepancy in phenotype severity may be a species-dependent difference, or a result of differences in knockdown vs. knockout approaches similar what we observed in our zebrafish studies.

Taken together, our results suggest an indispensable role of REEP5 in the SR/ER to maintain a highly differentiated SR network in cardiac muscle cells. In conclusion, we believe a more complete understanding of the role of REEP5 in the heart will be critical to understand the normal function of healthy versus diseased SR/ER.

## Methods

**Human myocardium.** All procedures involving human samples were approved by the University Health Network Ethics Review Committee (REB reference #158781-AE, 136800-AE, and 10-0703) and were performed in accordance with the guidelines. Written informed consent was obtained from all patients for use of their samples in research prior to study commencement. Human idiopathic tissue was isolated from a 58-year-old female and the ischemic cardiomyopathy tissue was isolated from a 68-year-old male during implantation of a left ventricular assist device at the Peter Munk Cardiac Centre (Toronto, Ontario). Healthy human left ventricle lysates were obtained from BioChain (Cat#: P1234138).

**Mouse heart disease tissues.** In all, 4 and 8 week TAC cardiac tissue were collected as described[53]. Two-week post LAD induced MI hearts were induced and isolated as described previously[54]. Finally, 16-week-old PLN R9C mutant mice, a model of human DCM were analyzed as described[55].

**Adult and neonatal mouse cardiac myocyte isolation.** All experimental procedures were conducted in accordance with the Animal Care Guidelines approved by University of Toronto Animal Use and Care Committee. Protocol for isolation of adult mouse cardiac myocytes was modified from Ackers-Johnson et al.[56]. Briefly,

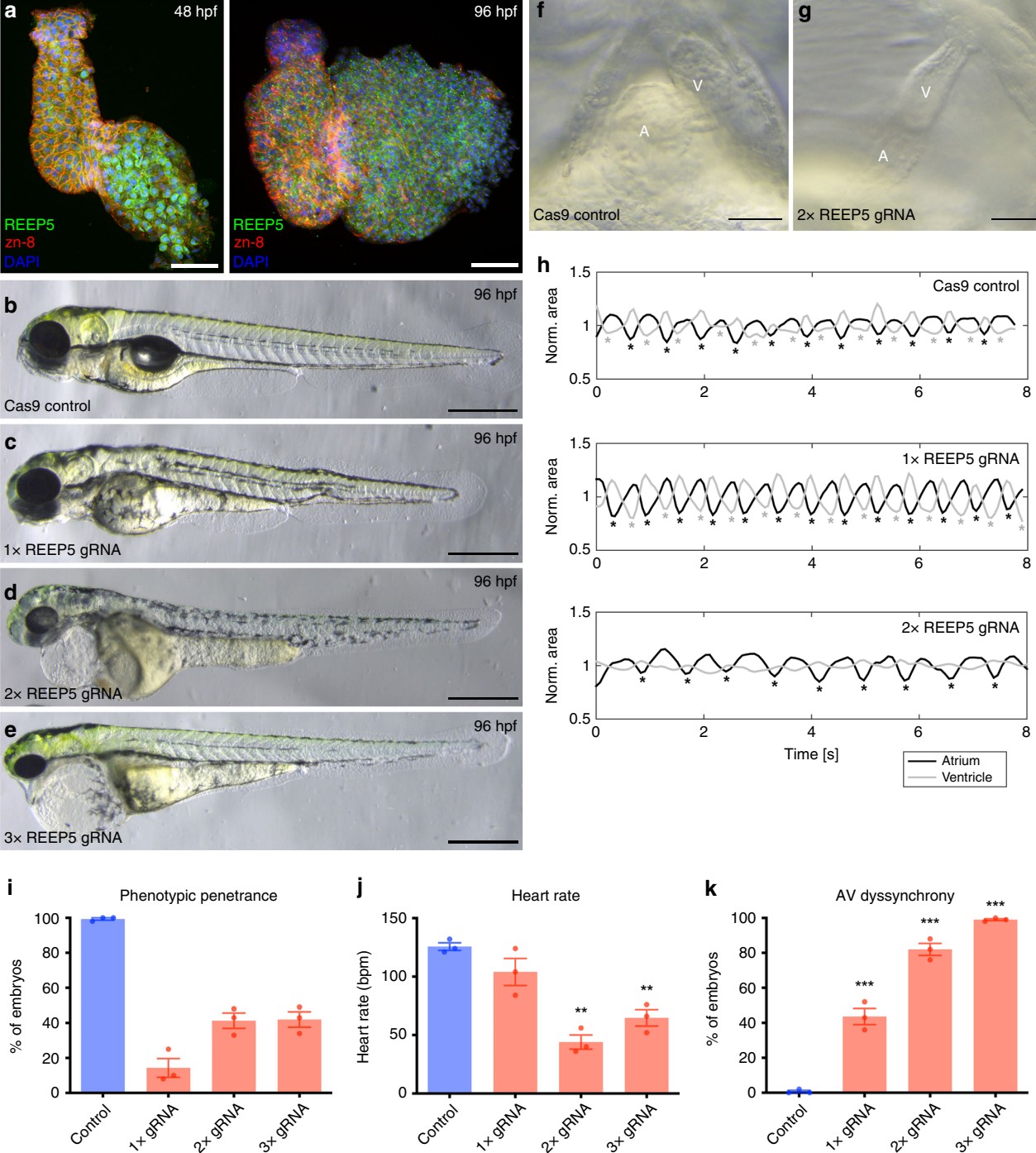

**Fig. 7 In vivo CRISPR/Cas9-mediated REEP5 depletion in zebrafish embryos leads to cardiac abnormalities. a** Immunofluorescence analysis of REEP5 expression in wild-type zebrafish hearts at 48 and 96 hpf. Co-staining for REEP5 (green), zn-8 (general plasma membrane; red) and DAPI (nuclear; blue) revealed REEP5 expression in both ventricular and atrial cardiac myocytes, with no nuclear or membrane localization evident. Scale, 50 μm. **b-e** CRISPR/ Cas9-mediated depletion of REEP5 with varying gRNA concentrations reveals the contribution of REEP5 to embryonic heart development in zebrafish embryos. Images shown are representative of 331 control, 410 2× REEP5 gRNA, and 296 3× REEP5 gRNA total injected embryos, n = 3 independent biological replicates. Scale, 500 μm. **f, g** Optical imaging analysis of control and 2× REEP5 gRNA injected embryos shows cardiac looping defects associated with REEP5-targeted gRNA injection. Scale, 10 μm. **h** Movie analysis of REEP5 gRNA injected embryos shows dyssynchronous atrio-ventricular beating rhythms compared to control zebrafish embryos. Area profiles were smoothened with a Gaussian curve with σ = 0.2 s. **i-k** Bar graphs showing **i** phenotypic penetrance represented as percentage, **j** heart rate represented as beats per minute (bmp), and **k** atrioventricular beating dyssynchrony represented as percentage from 331 control, 410 2× REEP5 gRNA, and 296 3× REEP5 gRNA total injected embryos, n = 3 independent biological replicates. Asterisks indicate a statistically significant p value in a Tukey's multiple comparison analysis where *p < 0.05, **p < 0.01, and ***p < 0.001; data are presented as mean ± SEM.

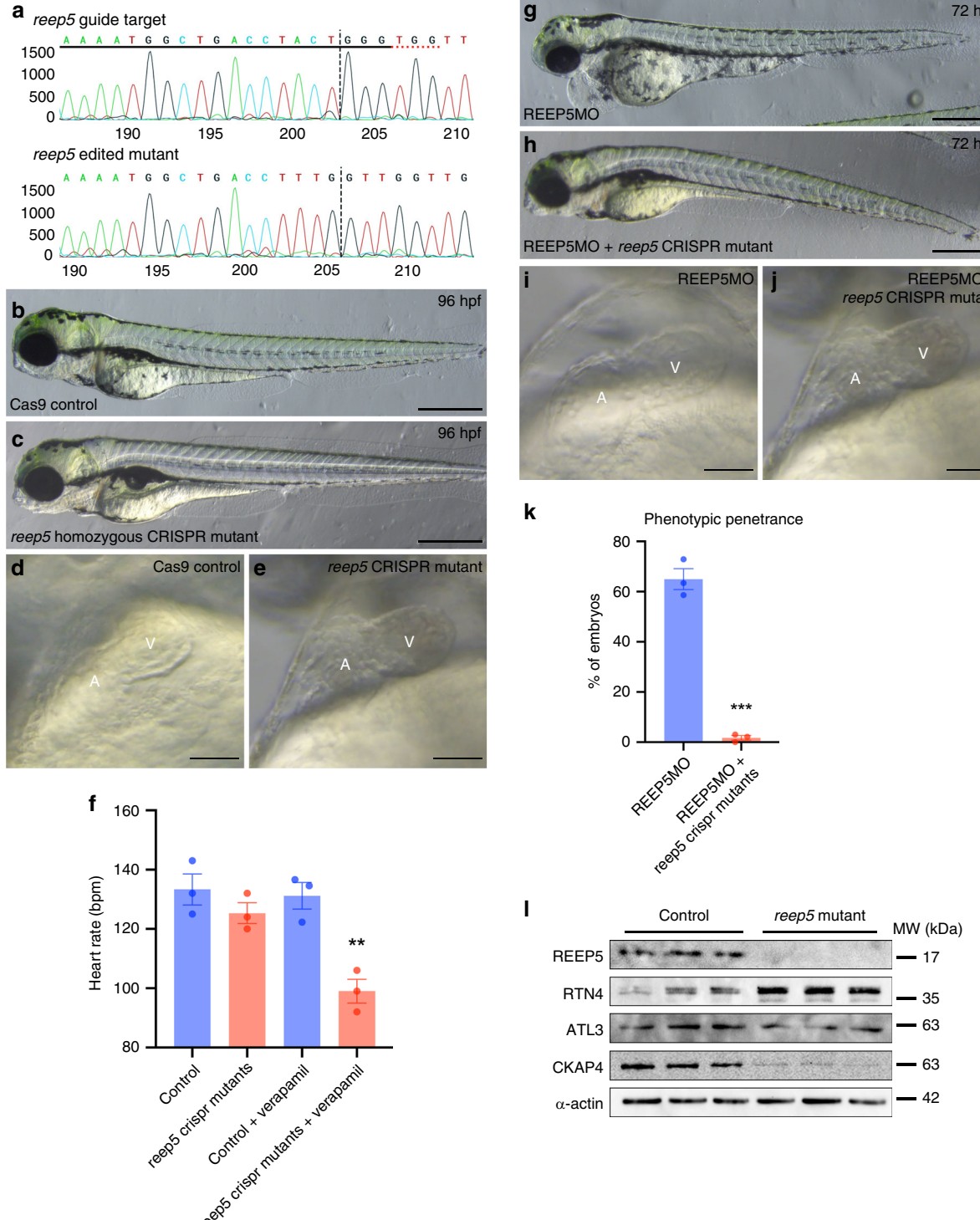

6–8-week-old male CD1 (Jackson Labs) mouse hearts were perfusion-isolated using 10 mL of EDTA buffer, followed by digestion with warmed collagenase II buffer (525 units/mL; LS004176, Worthington Biochemical Corp., Lakewood, NJ, USA). Following dissociation and washing, myocytes were resuspended in culture media and plated on laminin coated glass-bottom dishes. Primary mouse neonatal cardiac myocytes were isolated from CD1 mice as previously described[57]. Pups (p1–3) were decapitated and hearts were washed in ice-cold Hanks-balanced salt solution without calcium and magnesium. Atria were discarded and ventricles were dissected into 2-mm fragments, and enzymatically digested in 0.05% trypsin (Hank's balanced salt solution) on an end-to-end rotor overnight at 4 °C. Next day, cells were released with 450 unit/mL collagenase II (LS004176, Worthington Biochemical Corp., Lakewood, NJ) at 37 °C (15 min/digestion) until no tissue remaining. Collected cells were filtered through a 70-μm cell strainer (08-771-2, Fisher

Scientific), span down at 200*g* for 5 min at room temperature and pre-plated at 37 °C for 2 h in DMEM/F12 medium supplemented with 10% fetal bovine serum (FBS)/horse serum. Cells were then seeded on gelatin-coated glass-bottom dishes for 48 h before being maintained in culture medium (insulin-transferrin-sodium selenite media supplement, 10 μM BrdU, 20 μM cytosine β-d-arabinofuranoside, 2.5 μg/mL sodium ascorbate, 1 nM LiCl, 1 nM thyroxine).

**Cell culture**. Human embryonic kidney (HEK-293T) cells and murine C₂C₁₂ myoblasts were maintained in DMEM supplemented with 10% FBS, penicillin and streptomycin. For transgene expression, HEK-293T cells were transfected using polyethylenimine (Polysciences, Inc., Warrington, PA) and C₂C₁₂ myoblasts were transfected with Lipofectamine 2000 (Invitrogen) as described[57].

**Fig. 8 Genetic compensation in *reep5* CRISPR knockout zebrafish mutant. a** Sequencing analysis generated by Sanger Synthego shows *reep5* gRNA targeted and edited homozygous CRISPR mutant sequences in the region around the guide sequence. The horizontal black underlined region represents the guide sequence and the horizontal red underline indicates the protospacer adjacent motif (PAM) site with the vertical black line representing the cut site. **b, c** CRISPR/Cas9-mediated *reep5* homozygous mutant compared to control zebrafish embryos, $n = 3$ independent biological crosses. Scale, 500 μm. **d, e** Optical imaging analysis of control and *reep5* CRISPR mutant embryos shows normal cardiac chamber orientation and development in the *reep5* CRISPR mutant hearts. Scale, 10 μm. **f** Heart rate analysis of control and *reep5* homozygous mutant embryos upon 600 μM verapamil treatment for 30 min; a total of 46 control, 24 *reep5* CRISPR mutants, 32 control + verapamil, and 36 *reep5* CRISPR mutants + verapamil embryos were analyzed, $n = 3$ independent biological crosses; data are presented as mean ± SEM. **g, h** *reep5* CRISPR mutants injected with REEP5 MO (1 ng) appears phenotypically normal compared to overt cardiac abnormalities in the REEP5 morphant embryos. Images shown are representative of 287 REEP5 MO and 232 *reep5* CRISPR mutants injected with REEP5 MO embryos, $n = 3$ independent biological crosses. Scale, 500 μm. **i, j** Optical imaging analysis of REEP5 MO and *reep5* CRISPR mutants injected with REEP5 MO embryo hearts shows normal cardiac chamber orientation and development in the *reep5* CRISPR mutant hearts. Scale, 10 μm. **k** Bar graph showing phenotypic penetrance represented as percentage from 287 REEP5 MO and 232 *reep5* CRISPR mutants injected with REEP5 MO embryos, $n = 3$ independent biological crosses; data are presented as mean ± SEM. **l** Immunoblot analysis of cardiac-enriched ER structure proteins (RTN4, ATL3, and CKAP4) in *reep5* CRISPR homozygous mutants shows upregulation of RTN4 in *reep5* CRISPR mutant hearts. Source data containing original uncropped immunoblots are provided as a Source Data file. Asterisks indicate a statistically significant $p$ value in a Tukey's multiple comparison analysis where $**p < 0.01$ and $***p < 0.001$.

**Membrane yeast two-hybrid experiments**. The MYTH system[58] was used to validate REEP5 membrane topology as previously described[27]. Briefly, N-terminal TF transcription factor tagged TF-Cub:REEP5 "bait" REEP5 protein and ubiquitin fragments tagged (NubI and NubG) "prey" proteins were tested for protein–protein interactions. Upon reconstitution of TF-Cub and NubI tagged membrane proteins, TF is released to enable reporter activation. A mutant form of the ubiquitin fragment (NubG) that contains a I13G mutation only allows reconstitution with Cub when brought in close proximity by physically interacting proteins. If the bait REEP5 protein is targeted to the membrane, the N-terminal tagged bait will expose its Cub fragment in the cytosol to interact with interact with NubI, but not with NubG. Both Cub and Nub tagged proteins must be exposed in the cytosol for reconstitution and subsequent reporter activation. Activation of the reporter system was measured as growth on media.

**Lentiviral transduction of isolated mouse cardiac myocytes**. Lentivirus-mediated transduction for shRNA delivery in mouse neonatal and adult cardiac myocytes was conducted as previously described[57]. Briefly, lentivectors were created by triple transfections of pLKO.1-mREEP5 (trcn0000106187, RNAi consortium), pSPAX2, and pMD2.G in HEK-293T cells. A lentivector carrying scrambled shRNA was used as control. For transduction of cardiac myocytes, mouse neonatal and adult cardiac myocytes were incubated with 8 μg/mL polybrene for 90 min and incubated with lentiviral solution for 21 h at 37 °C, 5% $CO_2$. Transduced cells were cultured in fresh media for at least an additional 24 h prior to any experiments.

**Adeno-associated viral transduction in vivo in mice**. All adeno-associated viral (AAV9) parent vectors were obtained from Dr. Roger Hajjar (Mount Sinai School of Medicine, New York) and high titer AAV9 vectors were generated via a triple transfection approach in HEK293 cells as previously described[59,60]. Briefly, HEK293 cells were transfected with 15 μg of pDG9 plasmid and 5 μg desired AAV plasmid per plate. HEK293 cells were harvested 48 h post transfection for subsequent purification by filtration and iodixanol. For in vivo transduction, neonatal (P10) pups from wild-type CD1 mice (Jackson Labs) were injected intraperitonially $5 \times 10^{11}$ genomic titer of AAV9 virus carrying either mouse REEP5 shRNA or scrambled shRNA sequence (non-expressing control) and were monitored daily.

**Echocardiographic assessment of cardiac function in mice**. All mice were anesthetized with 1.5% isoflurane and placed on an animal warming pad maintained at 37 °C throughout the procedure. Heart rate, respiration, and body temperature were monitored and maintained consistently throughout the ultrasound imaging session. All mice were imaged in the supine position using Vevo 3100 Imaging system (Visual Sonics Inc., Toronto) with a 30 MHz probe. Standard 2D echocardiographic assessment (B-Mode and M-Mode) was performed through the left para-sternal acoustical window to assess left ventricular dimensions and heart function[61,62]. M-mode recording from the middle segment of the left ventricle was made to measure the left ventricular anterior and posterior wall thickness at peak systole and end diastole. Pulsed Wave Doppler Mode was used to acquire doppler flow profiles and to record velocity time integrals. Subsequent measurements and analyses were performed using the Vevo 3100 2D quantification software to calculate interventricular septal thicknesses, posterior wall thickness, systolic dimension, ejection fraction, and cardiac output.

**Immunoblotting and IF**. Protein lysates from cells and tissues were harvested in radioimmunoprecipitation assay buffer (RIPA, 50 mM Tris-HCl; pH 7.4, 1% NP-40, 0.5% sodium deoxychloate, 0.1% SDS, 150 mM NaCl, 2 mM EDTA), supplemented with protease and phosphatase inhibitors (Roche), for 30 min on ice, spun

down at 15,000g/4 °C. Soluble fractions were saved for immunoblotting. For live imaging, cells were seeded on glass-bottom dishes (MatTek Corp. Ashland, MA), and transfected with plasmids for fluorescence protein expression. Cells were directly visualized 24 and 48 h following transfection. For IF, cells were fixed with 4% paraformaldehyde for 10 min on ice, permeabilized with ice cold 90% methanol at −20 °C for 10 min, and blocked with blocking solution (3% FBS and 0.1% TritonX-100 in PBS) at room temperature for 1 h. Primary antibody staining was carried out overnight at 4 °C and fluorophore-conjugated secondary antibody staining was performed at room temperature for 1 h in the dark. Samples were visualized with Zeiss spinning disk confocal microscopy (Zeiss Spinning Disk Confocal Microscope). Three-dimensional reconstruction of z-stack images was carried out using Imaris 8.1 software (Bitplane, Switzerland).

**Transmission electron microscopy**. Isolated adult mouse cardiac myocytes were fixed in 2.5% glutaraldehyde in 0.1 M phosphate buffer, while zebrafish were fixed in 2% paraformaldehyde in 0.1 M sodium cacodylate at 4 °C overnight. Samples were postfixed in 1% osmium tetroxide buffer and processed through graded alcohols and embedded in Quetol-Spurr resin. Sections of 90–100 nm were cut and stained with uranyl acetate and lead citrate and imaged at ×20,000 magnification using a Hitachi TEM microscope at the Department of Pathology, St. Michael's Hospital (Toronto, Canada) or a FEI Tecnai 20 TEM microscope at the Hospital for Sick Children (Toronto, Canada)[35].

**Antibodies**. Primary rabbit polyclonal anti-REEP5 antibody (IB: 1:1000 dilution, IF: 1:800 dilution; 14643-1-AP; Proteintech), polyclonal anti-Nogo-A/B (IB: 1:1000 dilution, IF: 1:1000 dilution; PA1-41220; ThermoFisher), polyclonal anti-ATL3 antibody (IB: 1:1000 dilution, IF: 1:800 dilution; PA5-24652; ThermoFisher), polyclonal anti-CKAP4 antibody (IB: 1:1000 dilution, IF: 1:800 dilution; PA5-42926; ThermoFisher) and mouse monoclonal anti-Alpha sarcomeric actinin antibody (IF: 1:500 dilution; MA1-22863; ThermoFisher), monoclonal anti-Triadin antibody (IF: 1:500 dilution; MA3-927; ThermoFisher), monoclonal anti-RyR2 antibody (IF: 1:500 dilution; ab2827; Abcam) were used for immunocytochemistry and immunoblot studies. Anti-alpha tubulin (#2144), anti-GRp78 (#3177), anti-GRp94 (#2104), anti-GAPDH (#2118) antibodies from Cell Signaling; anti-ATF4 (ab216839), anti-Caspase12 (ab62484) antibodies from Abcam were used for immunoblot studies at 1:1000 dilution. Anti-GFP (sc390394) antibody from Santa Cruz Biotechnology was used for immunoblot studies at 1:1000 dilution.

**ROS and MMP measurements**. Intracellular ROS was measured using CellROX Deep Red reagent (Invitrogen) according to the manufacturer's instructions. Neonatal cardiac myocytes were cultured and seeded in 96-well plates or glass-bottom dishes. Cells were incubated with the ROS scavenger, Tiron (1 mM, Sigma) for 30 min at 37 °C, 5% $CO_2$ followed by a 30 min incubation with CellROX reagent at a final concentration of 5 μM. MMP was detected in cells using 50 nmol/L of vital mitochondrial dye JC-1 (Molecular Probes). Cells were washed with culture medium before and after a 20 min incubation period with JC-1 at 37 °C, 5% $CO_2$; fluorescence was then measured via plate-reader analysis[35].

**MTT viability assay**. Cell viability was assessed by performing 3-(4,5-dimethyl-thiazol-2-yl)-2,5-diphenyltetrazolium bromide (MTT) assay (Sigma-Aldrich) according to the manufacturer's instructions[61]. Neonatal cardiac myocytes were seeded in 96-well plates. Cells were incubated in absence of light with 0.5 mg/ml MTT at 37 °C, 5% $CO_2$ for 1 h. Untransformed MTT was removed and formazan crystals were solubilized with dimethyl sulfoxide (DMSO) followed by fluorescence measurement at 570 nm using Perkin Elmer plate reader.

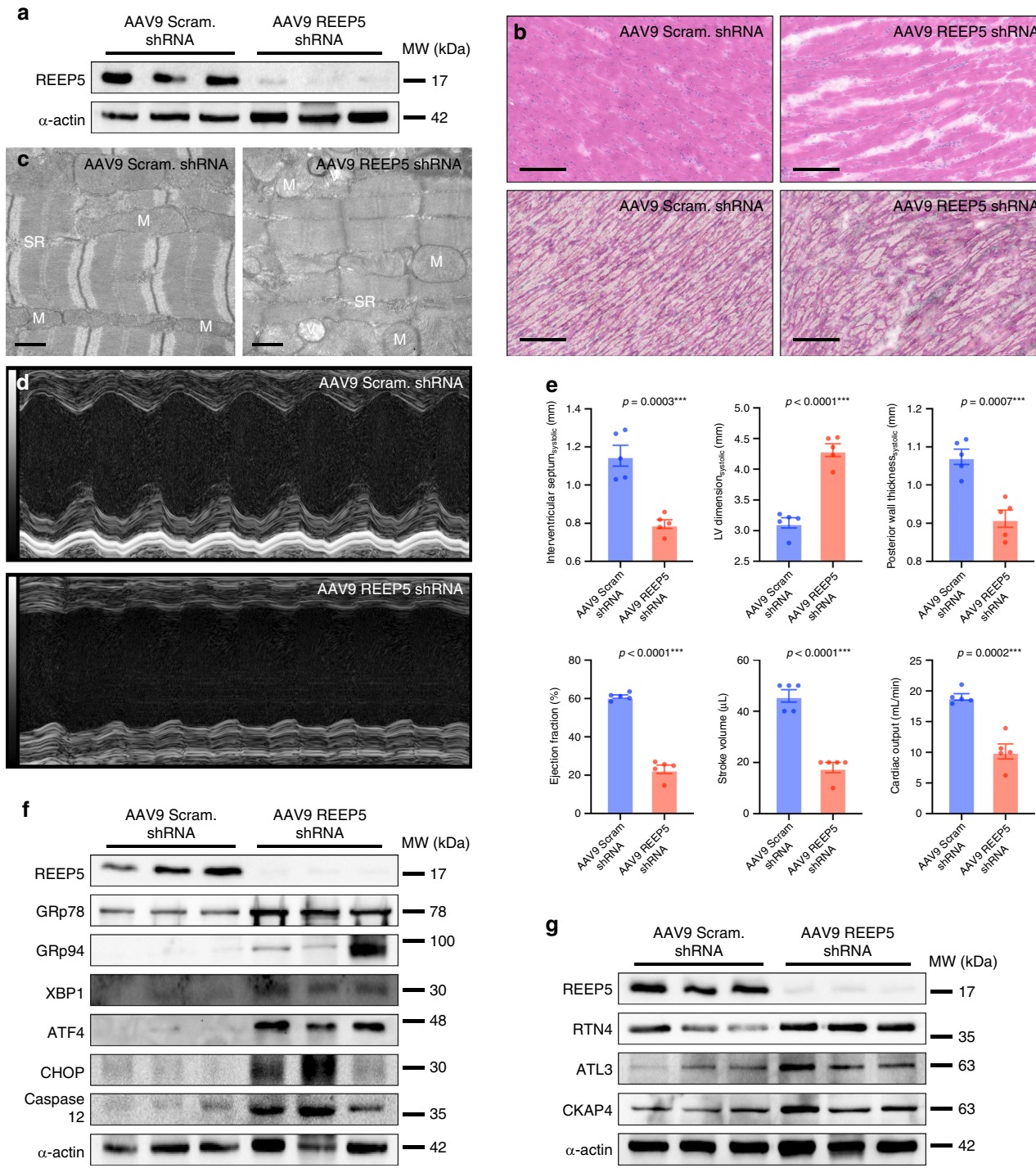

**Fig. 9 In vivo AAV9-mediated REEP5 depletion in mice results in increased cardiac fibrosis, activated cardiac ER stress, cardiac dysfunction, and death. a** Immunoblot analysis of AAV9-mediated REEP5 depletion in mice 4 weeks post viral infection. **b** Histological analyses of AAV9 Scram shRNA and REEP5 shRNA injected mouse hearts at 4 weeks post viral infection. Top panels: H&E, bottom panels: Masson's trichrome stain. Scale, 20 μm. All images shown are representative of approximately 20 total images captured per condition, *n* = 3 independent biological replicates. **c** Transmission electron microscopy analysis of AAV9 REEP5 shRNA-injected myocardium reveals degeneration of muscle fibers and disrupted SR membranes and organization compared to scram controls. M mitochondria, SR sarcoplasmic reticulum, V vacuoles. Scale, 0.5 nm. All images shown are representative of approximately 20 total images captured per condition, *n* = 3 independent biological replicates. **d** In vivo echocardiographic heart function assessment of AAV9 scram and AAV9 REEP5 shRNA-injected mice at 4 weeks post viral infection, *n* = 5. **e** Echocardiographic M-mode and B-mode measurements showed significantly compromised cardiac function with reduced cardiac ejection fraction and cardiac output, *n* = 5 independent biological replicates. Asterisks indicate a statistically significant *p* value in a Tukey's multiple comparison analysis where \*\*\**p* < 0.001; data are presented as mean ± SEM. **f** Immunoblotting analysis of cardiac ER stress markers (GRp78, GRp94, XBP1, ATF4, and CHOP) and ER-dependent apoptosis (caspase 12) upon AAV9-induced REEP5 depletion in vivo in mice, *n* = 3; *p* values are shown. **g** Immunoblots of RTN4, ATL3, and CKAP4 upon AAV9-induced REEP5 depletion in vivo in mice, *n* = 3 independent biological replicates. Source data containing original uncropped immunoblots are provided as a Source Data file.

**Protein purification and mass spectrometry analysis**. Eluted REEP5 pulldown lysates were concentrated using a 3000 MWCO centrifugal concentrator (Millipore), reduced with 5 mM DTT for 30 min at 55 °C, alkylated with 25 mM iodoacetamide for 30 min at room temperature in the dark. Proteins were trypsinized using 5 μg Trypsin Gold (Promega) overnight at 37 °C and halted via acidification with formic acid to a concentration of 1%. Peptides were then passed through using OMIX C18 Tips (Agilent) and quantified via BCA assay (ThermoFisher). For REEP5-transfected lysates and untransfected controls, three biological replicates were analyzed on a Q Exactive Plus mass spectrometer (ThermoFisher). Briefly, 2 μg of peptides were separated on a 2-h 5–35% acetonitrile gradient on a 10 cm long, 75 μm i.d analytical column packed with 3 μm 100 Å C18 resin (Phenomenex). Data were acquired at $R = 70,000$ at 400 m/z with one full MS1 scan from 400 to 1500 m/z followed by 12 data-dependent MS2 scans at $R = 17,500$ with dynamic exclusion set to 60 s. Two biological replicates for each condition was run in technical duplicates. RAW files were searched using MaxQuant version 1.5.4.1[63] using the UniProt human proteome sequence database. Variable modifications were set for methionine oxidation and protein amino terminal acetylation. False-discovery rate was set to 1% using a target-decoy strategy. Proteins identified in only 2 of 8 MS experiments were excluded for comparison. Expression levels were used using iBAQ values determined by MaxQuant.

**In vivo studies in zebrafish**. Adult zebrafish were maintained according to Canadian Council on Animal Care (CCAC) and The Hospital for Sick Children Animal Service (LAS) guidelines. Transgenic zebrafish line used in this study was $myl7:EGFP^{twu34,64}$. Zebrafish embryos were grown at 28.5 °C in embryo medium as previously described[65]. Standard techniques were used for microinjection[66]. Microinjection was repeated a minimum of five times with consistent results. Morpholino oligos were purchased from Genetools (Oregon, USA). The ATG REEP5 morpholino (5′-GCC GCC ATG ATT TGT CTG AAG GGA T-3′) and ATG REEP6 morpholino (5′-AGT CAC CAT GTT TGC CAT ATT TAC A-3′) were injected at a concentration of 1 ng/nl at the 1-cell stage of development. Bright-field images were taken using a Zeiss AXIO Zoom V16. Whole-mount immunofluorescence (IF) was carried out as previously described[67]. Secondary reagents AF488 goat-anti-mouse IgG1 recognizes monoclonal antibody S46 (atrial myosin heavy chain), AF555 goat anti-mouse IgG2b recognizes monoclonal antibody MF20 (Myosin heavy chain), AF488 goat anti-rabbit IgG recognizes antibody REEP5 and goat anti-mouse IgG 568 recognizes monoclonal neuronal cell surface marker (zn-8). MF20, S46 and ZN-8 primary antibodies were obtained from the Developmental Studies Hybridoma Bank (University of Iowa, IA, USA). Immunoblotting for REEP5 detection was carried as described above using 40 embryos per experimental condition. IF confocal images were taken with a Nikon A1R laser scanning confocal microscope or using Zeiss spinning disk confocal microscopy (Zeiss LSM 510/AxioImager.M1 Confocal Microscope). For in vivo RNA rescue experiments, Full-length wild-type zebrafish reep5 coding sequence was amplified from 48 hpf wild-type zebrafish cDNA. The coding sequence was modified to avoid interaction with the antisense morpholino while producing the same amino acids (5′ATGGCAGCAGCGTTA...TAA-3′). This modified coding sequence was subcloned into the pCS2+ vector. For mRNA production, the pCS2+ :reep5 construct was linearized via NotI digestion and in vitro transcription was carried out using the mMessage mMachine SP6 kit (Ambion). To validate the morphant phenotype, RNA rescue was performed via subsequent injection of antisense reep5 morpholino and reep5 mRNA. 1 ng/μL (1 nL volume) of reep5 morpholino was injected at the one-cell stage, immediately followed 100 pg (1 nL volume) of reep5 mRNA. Embryos were incubated at 28.5 °C and monitored daily.

CRISPR/Cas9 reep5 mutants were generated as previously described[68]. To generate reep5 CRISPR mutants, exon sequences were analyzed for optimal cut sites using the Benchling web-based service (www.benchling.com). gRNA target sequences were ranked by highest off-target score followed by highest on-target score which resulted in a gRNA cut site in exon 3 targeting the beginning of the second transmembrane domain (5′-AAAATGGCTGACCTACTGGG). The REEP5-specific oligo using the T7 promoter with the standard overlap region was ordered from ThermoFisher (5′-TTAATACGACTCACTATAGG-**AAAATGG CTGACCTACTGGG**-GTTTTAGAGCTAGA). Once the template oligo was assembled, transcription was performed using the T7 Quick High-Yield RNA Synthesis Kit (NEB). The resulting RNA was purified using the standard ethanol/ammonium acetate protocol[69]. An injection mix was prepared with gRNA and Cas9 protein in a 1:1 ratio at a concentration of 2 mM (1×). 1–3 nL of the solution was injected into zebrafish embryos at the 1-cell stage of development to achieve 1×, 2×, or 3× amounts of gRNA. Standard microinjection techniques were used[69]. CRISPR/Cas9 gRNA cutting was evaluated in injected and uninjected 4 dpf embryos using a LightCycler 96 from Roche. Analysis was conducted using the Roche template for the HRM assay. Injections and HRM analysis were repeated 3 times with consistent results. Germline CRISPR reep5 homozygous MZ mutants were generated by performing an additional cross with an identified homozygous female reep5 mutant to rule out maternally-contributed mRNA expression of REEP5. The effects of verapamil were tested in 2 dpf wild-type and reep5 homozygous mutant fish. Concentrations of 10, 5, 2.5, 1.25, and 0.6 mg/mL of verapamil in 1 mL of E3 medium (5 mM NaCl, 0.17 mM KCl, 0.33 mM CaCl₂, 0.33 mM MgSO₄) were tested, with 0.6 mg/mL having no measurable effect on

wild-type embryo heartbeat. Embryos were examined for phenotypes following 30 min of drug exposure using a Zeiss standard dissection microscope. Heart rate was recorded before and after drug treatment with atrial and ventricular heartbeats counted separately over a period of 15 s each.

**Plasmids and reagents**. Human REEP5 (NM_005669.4) was subcloned into pLD-puro-CnVA[38] and pcDNA-DEST40 (V5/6×-His epitope) plasmids by Gateway cloning technology according to manufacturer's manual (Invitrogen). Truncation deletions were generated using available restriction digest sites. For the truncation deletion removing the cytosolic C-terminus region, only sequences around amino acid 114 were viable restriction sites, resulting in the 1–114 and Δ114–189 constructs; attempts at generating different C-terminal mutants were not successful. pmCherry-Sec61β was created as previously described[70]. Briefly, mouse Sec61β (NM_024171) was PCR amplified with primers 5′-CAT CAT AGA TCT ATG CCG GGT CCA ACG CCC-3′ (mSec61 SP, BglII) and 5′-GTA GTA GAA TTC CTA TGA TGA TCG CGT GTA CTT GCC CCA-3′ (mSec61 AP, EcoRI) and subcloned in a pmCherry-C1 plasmid with the same restriction sites. The PCR was performed with following parameters: 1 cycle of 98 °C for 30 s, 35 cycles of 98 °C for 10 s and 72 °C for 10 s, and 1 cycle of 72 °C for 2 min. All constructs were validated by sequencing at the ACGT Corp (Toronto, ON, Canada).

**Movie analysis**. Zebrafish bright-field movies were captured using a Zeiss AXIO Zoom_V16 at ×63 magnification. Embryos injected with or without REEP5 MO were subjected to PTU for removal of pigmentation prior to movie capture. To measure atrial and ventricular area, we performed imaging analysis using SIESTA, a custom-built image analysis package[71,72]. Specifically, we applied the LiveWire algorithm implemented in SIESTA for semi-automated delineation of heart chambers. Area profiles were smoothened with a Gaussian curve where $\sigma = 0.1$ s and normalized by dividing by the mean area in each profile.

Adult mouse cardiac myocytes time-lapse images were obtained using a Revolution XD spinning disk confocal microscope equipped with an iXon Ultra 897 camera (Andor, Belfast, UK) and a ×1.5 coupling lens. A ×40 oil immersion lens (Olympus, NA 1.35) was used to acquire one 16-bit image every 40 ms. Image analysis was performed using algorithms developed with Matlab (MathWorks, Natick, MA) and DIPImage (Delft University of Technology, Delft, Netherlands). To measure cardiac myocyte area over time, we binarized images of isolated cardiac myocytes using the image mean as the threshold. A Savitzky-Golay filter with second order polynomials and a window size of 75 pixels was used to smooth the cardiac myocyte outlines prior to quantification. To measure fractional shortening, we calculated the equation of the line along the longest axis of each cardiac myocyte and measured the length of the cell under the line over time. We normalized length data to the first time point at which the cell is at rest between contractions, and detrended the signal by subtracting the best-fit sixth order polynomial. We considered a contraction event to correspond to fractional shortening of 5% or more of the initial rest length and the findpeaks function in Matlab was used to identify contractile pulses and averaged their amplitude for each cell. Frequency was calculated as the inverse of the mean period between contractile events.

**Statistical analyses and reproducibility**. Experimental measurements were analyzed and graphically presented by the GraphPad Prism8 software. Descriptive statistics are shown as mean ± SEM. Data are plotted as mean ± SEM with individual data points as scatter plot overlays. Normally distributed data were analyzed using two-way ANOVA followed by post hoc Tukey's multiple comparison test for each mean comparison. Experimental mean-fold protein intensities, relative to controls from triplicate assays, were considered different from controls at the $p < 0.05$ significance level. All immunoblots shown are representative immunoblots from a minimum of three independent biological replicates. Source data containing the raw data underlying all reported averages in graphs and charts and all original uncropped immunoblots are provided as a Source Data file.

**Reporting summary**. Further information on research design is available in the Nature Research Reporting Summary linked to this article.

## Data availability
The authors declare that all supporting data are available within the article and supplementary files, or from the corresponding author upon reasonable request.

Publically available RNASeq and proteomic data were downloaded and analyzed. Specifically, the following mRNA Affymetrix transcript data from GEO profiles in NCBI for mouse cardiac tissue and disease were obtained: severe DCM (GDS487/96115_at), 2d, 10d, 21d TAC (GDS794/103886_at), MI (GDS3655/14195), alpha-tropomyosin mild HCM (GDS2134/1419398_a_at), and Emery-Dreifuss (GDS2884/1426376_at). Similarly, we obtained human mRNA Affymetrix GEO profiles for human heart transplant rejection (GDS2386/208872_s_at), human inflammatory DCM (GDS2154/208873_s_at), DCM (GDS4772/8113542) as well as human idiopathic and ischemic CM (GDS651/208872_s_at). Data were reported as normalized hybridization signals. Comprehensive human RNASeq based transcript levels were obtained from the Human Protein Atlas Project[73]. For that normal human tissue, RNA samples were extracted from frozen tissue

sections in the Uppsala Biobank. Data were reported as the abundance in "Transcript Per Million" (TPM) as the sum of the TPM values of all its protein-coding transcripts[73].

The source data underlying Figs. 1e, h, 3b–e, g, 4d, e, 6c, 7i–k, 8f, k, 9e, and Supplementary Fig. 3a are provided as a Source Data file.

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

## Acknowledgements

We would like to thank Dr. Stephen Strittmatter (Yale University) for providing human Myc-Nogo-A and Myc-Nogo-B plasmids. We thank Aaron Wilson and Wenping Li for providing expert technical support; and Dr. Yu-Qing Zhou for providing expert experimental support for all in vivo echocardiographic experiments.

This project was funded by the Ted Rogers Centre for Heart Research Innovation Fund to A.O.G. and I.C.S.; the Heart and Stroke Richard Lewar Centre of Cardiovascular Excellence; and CIHR Award to A.O.G. (PJT-155921 and PJT-166118) and a Discovery Grant from NSERC (RGPIN-#05618). S.H. Lee was supported by a NSERC Postgraduate Scholarship. S.H. Lakmehsari was supported by a CGS-Master's Award. N.G. was supported by the Philip Witchel Research Fellowship. J.C.Y. was supported by an Ontario Graduate Scholarship.

## Author contributions

S.H. Lee, S.H. Lakmehsari, H.R.M., I.C.S., and A.O.G. designed the experiments, analyzed the data, and wrote the paper. H.R.M. and N.G. carried out the zebrafish experiments and contributed to the data analysis and paper writeup. S.H. Lee, V.W., and I.S. carried out the yeast-two-hybrid experiments and contributed to the data analysis and paper writeup; S.H. Lee, S.H. Lakmehsari, and T.M. performed and analyzed the imaging experiments. J.C. performed and analyzed the proteomics experiments. J.C.Y. and R.F.-G. provided the SIESTA software and custom scripts for image analysis of heart beats. M.M. and P.L. performed and analyzed the TAC experiments in mice. All authors (S.H. Lee, S.H. Lakmehsari, H.R.M., N.G., T.M., A.C.T.T., J.C.Y., M.M., V.W., P.L., R.F.-G., I.S., P.S., T.K., I.C.S., and A.O.G.) reviewed and approved the paper.

## Competing interests

The authors declare no competing interests.
