## [Peer Review File · Nature Communications]

Reviewers' Comments:

Reviewer #1:

Remarks to the Author:

Formation of the tubular ER is of great physiological significance. In this manuscript, Lee et al. showed that depletion of one of the ER tubule-forming proteins REEP5 causes ER stress in cells and heart development defects in zebrafish. Overall, the authors presented a big pile of data, but the purpose for many experiments are not clear. In addition, some experiments were poorly designed and some results were mistakenly interpreted. It is interesting to see ER stress and cardiac related phenotypes upon loss of REEP5. However, it is not clear whether these are specific functions of REEP5 in heart or in general, maintenance of ER morphology is important.

Major concerns:

1. At the beginning of the introduction, the authors try to provide an overview of what is known about ER shaping proteins. However, several references are inappropriate, and several are missing. For examples, very specific papers, instead of reviews, were chosen for general functions of the ER; ref #5 talks about a specific type of ER sheets, but was quoted for general description of the ER network; ATL and LNP were mentioned, but no ref was seen for ATL; and finally Climp63 was introduced but again with no proper refs. Similar issues appear throughout the manuscript. It is suggested that the authors read ER shaping-related literatures carefully and then redo the refs.
2. In Fig. 1, the authors invest a lot of efforts on dissecting the topology of REEP5. REEP5 is also known as DP1, the first well-characterized REEP in the family. In fact, similar experiments have been done in the paper (Voeltz GK, Cell 2006) from which DP1/REEP5 was identified as tubule-shaping protein. Unfortunately, such a crucial reference was not even in the reference list.
3. To justify their emphasis of REEP5 on cardiac related functions, the authors claim that REEP5 is the most abundant REEP in fetal heart and adult heart. However, it is not mentioned how the experiments were performed. In the legend of Fig. 1H, instead of providing such information, the authors went on to describe their reading of the results. Even if real-time PCR was performed to assess transcript abundance, it is not clear how results for different genes can be compared, assuming different primers were used.
4. In Fig. 1I, a commercially available antibody against REEP5 was used to determine its levels in various tissues. Three major bands were seen and were interpreted as monomer, dimer, and trimer. It has been seen before that these integral membrane proteins can sometimes form SDS-resistance oligomers, and these oligomers become more prominent when chemical cross-linkers were added. However, it is always the case that monomer is the most abundant, with less dimer and even less trimer. Here, some of the "dimer" or "trimer" bands are much more prominent than monomer in the same lane. It thus raises suspicion on the specificity of the antibody. Without better evidence to support the quality of the antibody, it would also be difficult to assess the IF experiments performed later.
5. In Fig. 2, the authors conclude that REEP5 localizes to j-SR, presumably a specific subdomain of the SR. However, from the images, it seems that REEP5 mostly co-localizes with common markers of the SR, such as SERCA and RyR, and exists in the entire SR. If the j-SR enrichment is critical for its proposed function, then images with better resolution is needed.
6. When REEP5 was depleted in cells, the authors noticed that ER stress was triggered. However, it is not clear whether ER morphology, in particular the tubular ER network is affected. In panel C, the vesiculation of the ER is likely the consequence of sustained ER stress. Nevertheless, based on the given images, it is difficult to judge the quality of the tubular ER network. If the tubular ER remains unchanged, the data would suggest a role of REEP5 in addition to curvature generation. Then what causes the ER stress upon loss of REEP5 remains elusive. If the tubular ER is converted to sheets, it is still not clear why ER stress is induced. Previous studies suggest that ER expansion

(usually more sheets) would alleviate ER stress instead of causing it. It is therefore necessary for the authors to carefully evaluate the impact of REEP5 depletion on the tubular ER network, and more importantly provide a plausible explanation for the observed phenotype.

7. The choice 114-189 truncation is problematic. For one thing, it is shown in Fig. 1G and 4G that the second TM hairpin ends at residue 124. I think it is a labeling mistake in those figures. Even if 144-189 deletion preserves the RHD domain, it is still very likely that conserved amphipathic helix is deleted (see ref #23). Therefore, the subsequent defects of this mutant would likely be caused by destabilization of the protein folding. In addition, the defects of dimerization that the authors dwell here are very likely artificial. Firstly, disulfide bridge is less likely formed in cytosolic fragments, as the cytosol contains a reduced environment. Secondly, in panel C, the gradual disappearance of dimer did not cause corresponding accumulation of monomer, making it difficult to reconcile the results.

8. The authors used the entire Fig. 5 to demonstrate that REEP5 interacts with other ER shaping proteins. Given that similar results have been shown when ATL was identified, it is not clear why these results are informative. One may argue that the interaction between REEP5 and Climp63 is unexpected. Even if the interaction is real, the relevance is not addressed.

9. In the final figure, the authors try to argue that there is certain expression correlation between ER shaping proteins. However, in panel B, even the loading controls are quite different from each other, it would thus be difficult to conclude that depletion of REEP5 actually causes decreased levels of Rtn4 and Climp63. Furthermore, changes in panels A and B look quite different.

10. Finally, the correlation between REEP5-related ER stress and cardiac phenotypes has not been tested. It would strengthen the conclusion if ER stress to be monitored in zebrafish with REEP5 depletion.

Minor concerns:

1. Line 362, Climp63 is not found in yeast.

2. Line 388-393, it is confusing here. Did the authors try to show that REEP5 and Rtn4 form oligomers or they interact with each other. Ref. #29 shows that upon cross-linking, Yop1p or Rtn1p forms SDS-resistant oligomerization.

Reviewer #2:

Remarks to the Author:

Lee et al. have discovered a new and previously unappreciated role for REEP5 in the tubular endoplasmic reticulum, cardiomyocyte and heart development. They use a variety of techniques to define this role and interactions. While they do not specifically determine the mechanism involved (future studies) this study does provide an important advance to the field.

1. My main concern is the statistical methods used. 1-way ANOVAs were used for normally distributed parameters. This method is appropriate when determining if a single treatment produces an effect; however, when comparing two or more treatments (e.g., shREEP5 vs. sh-scrambled) a 2-way ANOVA is more appropriate.

2. Fig. 1I: The authors state, "REEP5 levels were the highest in mouse ventricle", but do not provide quantification. They state, "the predominant form in the ventricle was the 17kDa monomer of REEP5" and 1 sentence later state "while the monomer and trimer was the prevailing form in the ventricle". Please correct this inconsistency and quantify REEP5 monomer, dimer and trimer levels.

Minor:

Fig 1H Figure Legend: The phrase, "ubiquitously detected in many ... tissues" is contradictory. Please remove "ubiquitously".

Reviewer #3:

Remarks to the Author:

The report by Lee, Hadipour-Lakmehsari, Gibb, et al. reports on the role of the ER protein REEP5 in cardiac tissue. The authors show REEP5 is expressed in cardiomyocytes and c2c12 myoblast cells. Loss of REEP5 or overexpression of a truncated mutant produced large vacuoles suggestive of a disorganization of ER structure. Morpholino antisense knockdown of REEP5 in zebrafish was employed to verify a role for this protein in cardiac function and development.

Overall, the paper is difficult to read. Many experiments are not adequately explained (for instance the membrane 2-hybrid) making it difficult to assess the data presented. The figure legends are likewise devoid of detail needed to understand what one is supposed to observe in many cases.

Although the title states loss of REEP5 causes ER dysfunction, this isn't directly investigated. There is no evidence to show a vacuolated ER isn't able to perform ER functions. The evidence for REEP5 in cardiac function and development comes down to inadequate morpholino studies in zebrafish.

Overall, I do not find this paper meets the quality I have come to expect from Nature Communications.

Some general comments:

1. In the introduction, the phylogeny of the REEP proteins is mentioned as background, but in the results section it is phrased as though this was determined in this study. This should be clarified.
2. There needs to be more detail about the membrane yeast-two hybrid so that a reader can assess what is in front of them. The text says Ost1-Nubl is the positive control, and yet interaction with the control is used as proof of a result? This is either poorly worded or something is incorrect. What is it you want me to see on the yeast plates that tells me the REEP5 N-terminus is cytoplasmic? What does 5-FOA tell you in this particular setup? A diagram would be helpful.
3. There are numerous Western blots throughout the paper without any quantification. Density quantification should be used at a minimum to backup the qualitative statements about protein regulation and levels.
4. In the Figure 1 legend – there is an asterisk to denote REEP5 expression in ubiquitous tissues. Did the authors mean to say REEP 5 is ubiquitous and the only REEP in cardiac tissues denoted with an asterisk?
5. Line 260 – the authors report that cell death is not causing the morphological changes in CMNCs where REEP5 is depleted, and yet on Line 279 the authors conclude ER-dependent cell death is occurring. How do the authors reconcile this?
6. In the immunolocalization studies, given the ER is almost ubiquitous throughout the cell, it's hard to evaluate where there is true co-localization (in the same place through interaction) versus co-compartmentalization since the ER is everywhere and the signal is going to overlap with other structures in the cell. Especially in Figure2 – this looks like they are expressed in similar domains, but have a lot of non-overlap as well. How meaningful is this? What happens to the sarcomere structure in TEM upon REEP5 depletion?
7. On this note, in Supplemental Figure 3 – what is different between "merge" and "overlay"? The overlay shows there is a lot of non-overlapping expression, in fact more non-overlap than overlap. In the end, I am convinced they are both expressed in the same cell, but not sure what the localization tells me beyond that.
8. I was specifically asked to comment on the zebrafish work in this manuscript. The authors use one ATG translation blocking morpholino to REEP5 to make claims about the role of this protein in cardiac function and development. Based on the standard of the zebrafish community, one morpholino is not sufficient due to the concern of off-target effects. It is good to see that the protein is depleted on the Western, but no rescue experiments are performed to determine what is off-target and what is not.

9. Morpholinos are notorious for causing developmental delays and off-target effects. Couple that with the fact that the cardiac system is particularly easy to perturb if development, in general, is affected, I find it hard to believe any of the cardiac function data as it stands. Given the ease of CRISPR-Cas9 technology, a knockout of REEP5 would be most helpful, and a targeted knockout in the heart would be ideal for these experiments to be meaningful.

10. Other issues with the zebrafish work – it is not possible to determine the looping status of the hearts comparing D to E or F to G. In K-N hearts are seen as delayed in the cone stage, or looped, but in J the authors claim hearts remain linear. Are they saying the hearts loop and then become linear over time? Or that some heart fail to loop and remain midline?

11. The immunofluorescence in the zebrafish heart is not informative. Higher resolution imaging would be required to say anything, and given the ubiquitous nature of the ER, it's not clear what can be obtained from this type of analysis as presented.

Reviewers' comments:

Reviewer #1 (Remarks to the Author):

1. At the beginning of the introduction, the authors try to provide an overview of what is known about ER shaping proteins. However, several references are inappropriate, and several are missing. For examples, very specific papers, instead of reviews, were chosen for general functions of the ER; ref #5 talks about a specific type of ER sheets, but was quoted for general description of the ER network; ATL and LNP were mentioned, but no ref was seen for ATL; and finally Climp63 was introduced but again with no proper refs. Similar issues appear throughout the manuscript. It is suggested that the authors read ER shaping-related literatures carefully and then redo the refs.

Response: We thank the reviewer for these suggestions and agree with these comments. We have made the appropriate inclusions to our references in our revised manuscript. We have ensured that we cite the critical references in the field including: Voeltz GK, Cell, 2006; Hu J, Science, 2008; Hu J, Cell, 2009; Orso G, Nature, 2009 & Powers RE, Nature, 2017 and edited the manuscript carefully to ensure appropriate citations throughout the manuscript.

2. In Fig. 1, the authors invest a lot of efforts on dissecting the topology of REEP5. REEP5 is also known as DP1, the first well-characterized REEP in the family. In fact, similar experiments have been done in the paper (Voeltz GK, Cell 2006) from which DP1/REEP5 was identified as tubule-shaping protein. Unfortunately, such a crucial reference was not even in the reference list.

Response: Our original use of the split-ubiquitin yeast 2 hybrid assays in Fig 1C was to identify critical mammalian protein interactions for membrane proteins. We ultimately made use of our mass spectrometry expertise for this purpose. However, a side benefit is that we employed the Y2H approach to verify the topology of REEP5. However, the omission of the Voeltz citation in our earlier submission was unfortunate. We have ensured that we credit this important study correctly in our revised manuscript.

3. To justify their emphasis of REEP5 on cardiac related functions, the authors claim that REEP5 is the most abundant REEP in fetal heart and adult heart. However, it is not mentioned how the experiments were performed. In the legend of Fig. 1H, instead of providing such information, the authors went on to describe their reading of the results. Even if real-time PCR was performed to assess transcript abundance, it is not clear how results for different genes can be compared, assuming different primers were used.

Response: We understand the reviewer's concern concerning the experimental description of these studies since that is of critical importance. In this revised manuscript, we have ensured to provide a clear and focused rationale and description of these experiments. As outlined in our revised manuscript, we utilized multiple sources of normalized RNA-Seq expression levels of members of the REEP family of proteins from publically available

databases of mouse and human tissues to determine REEP expression levels across different tissues (Lines 188-195; 207-223).

4. In Fig. 1I, a commercially available antibody against REEP5 was used to determine its levels in various tissues. Three major bands were seen and were interpreted as monomer, dimer, and trimer. It has been seen before that these integral membrane proteins can sometimes form SDS-resistance oligomers, and these oligomers become more prominent when chemical cross-linkers were added. However, it is always the case that monomer is the most abundant, with less dimer and even less trimer. Here, some of the “dimer” or “trimer” bands are much more prominent than monomer in the same lane. It thus raises suspicion on the specificity of the antibody. Without better evidence to support the quality of the antibody, it would also be difficult to assess the IF experiments performed later.

Response: We appreciate the reviewer’s comments and we completely agree with the reviewer. It is correct that DP1/Yop1 can form oligomers and that the monomeric form is always the most abundant in yeast and mammalian cells (Shibata Y, JBC, 2008). However, in our studies, immunoblot analysis of REEP5 was conducted in various mouse tissues in the same blot and under the same conditions. The exact molecular architecture of RHD oligomers remains unclear and it is not known whether higher molecular weight oligomers are required for proper localization and function in more complex and differentiated tissues. In addition, to further validate the specificity of the anti-REEP5 antibody used in our study, we performed immunodepletion assays using bacterially expressed and purified REEP5. Under these conditions the immunoblots showed no visible bands at the monomer, dimer, or trimer positions (Fig. 1G), further validating the specificity of the anti-REEP5 antibody. However, that notwithstanding, we have included a statement of limitations with any antibody-based study (Lines 202-205).

5. In Fig. 2, the authors conclude that REEP5 localizes to j-SR, presumably a specific subdomain of the SR. However, from the images, it seems that REEP5 mostly co-localizes with common markers of the SR, such as SERCA and RyR, and exists in the entire SR. If the j-SR enrichment is critical for its proposed function, then images with better resolution is needed.

Response: We appreciate the reviewer’s concern and agree that REEP5 expression is apparent throughout the entire SR. However, in our study, not only we have shown REEP5 localization to general SR domains via co-staining with SERCA2, we specifically showed co-localization between REEP5 and several known junctional SR markers (alpha actinin, RyR2, and Triadin) followed by detailed three-dimensional reconstructive co-localization analysis using Imaris (Supp. 3; Lines 266-280). We have toned down our earlier statements to appease this concern of the reviewer, and have revised our earlier statement to read that REEP5 is expressed along the SR and into the specialized junctional SR.

6. When REEP5 was depleted in cells, the authors noticed that ER stress was triggered. However, it is not clear whether ER morphology, in particular the tubular ER network is affected. In panel C, the vesiculation of the ER is likely the consequence of sustained ER stress. Nevertheless, based on the given images, it is difficult to judge the quality of the

tubular ER network. If the tubular ER remains unchanged, the data would suggest a role of REEP5 in addition to curvature generation. Then what causes the ER stress upon loss of REEP5 remains elusive. If the tubular ER is converted to sheets, it is still not clear why ER stress is induced. Previous studies suggest that ER expansion (usually more sheets) would alleviate ER stress instead of causing it. It is therefore necessary for the authors to carefully evaluate the impact of REEP5 depletion on the tubular ER network, and more importantly provide a plausible explanation for the observed phenotype.

Response: We understand the reviewer's concern concerning the role of REEP5 and ER stress. We agree that these are very interesting points to consider. In fact, we believe that it is entirely possible that prolonged ER stress is causing the morphological changes we have observed in our study. In fact, in our study, when we pharmacologically induced ER stress in cardiac myocytes using tunicamycin, we saw a marked decrease in REEP5 protein expression levels, further supporting our hypothesis that REEP5 plays a crucial role in regulating SR development and maintenance in the heart (Fig. 3G). Hence, we are confident that the changes to the SR/ER morphology we are seeing are specific to the role of REEP5 in cardiac muscle. We have added several sentences within the revised text to highlight this potential outcome based on the loss of REEP5 (Lines 634-654).

7. The choice 114-189 truncation is problematic. For one thing, it is shown in Fig. 1G and 4G that the second TM hairpin ends at residue 124. I think it is a labeling mistake in those figures. Even if 144-189 deletion preserves the RHD domain, it is still very likely that conserved amphipathic helix is deleted (see ref #23). Therefore, the subsequent defects of this mutant would likely be caused by destabilization of the protein folding. In addition, the defects of dimerization that the authors dwell here are very likely artificial. Firstly, disulfide bridge is less likely formed in cytosolic fragments, as the cytosol contains a reduced environment. Secondly, in panel C, the gradual disappearance of dimer did not cause corresponding accumulation of monomer, making it difficult to reconcile the results.

Response: We appreciate and agree with the reviewer's comments. First, the designation of amino acids according to our analysis is correctly proposed in the text: the end of the TM domain is amino acid 123. Unfortunately, cloning strategies to specifically generate the 124-189 truncation repeatedly failed, leaving only the 114-189 truncation as a viable alternative. We have included this important detail in our revised text. However, of course we agree that it is possible that deleting a portion of the protein could cause destabilization and misfolding of the protein. As well, we entirely agree that by deleting the conserved amphipathic helix domain, which has been shown to be essential for membrane, curvature stabilization (Brady JP et al., 2015, PNAS) would introduce vacuolization in the SR/ER membranes. In our revised manuscript, we have further explained the effects we observed could be due to the deletion of the conserved APH domain of REEP5 (Lines 574-579). In addition, although deleting a portion of the protein could potentially cause protein destabilization and misfolding, however, in our study, we were able to detect stable and robust expression when we introduced the REEP5(1-114) mutant through immunoblot and immunofluorescence analyses as well as ER localization of the REEP5 mutant as shown by co-localization with a fluorescence-tagged ER marker, KDEL (Fig. 5 E,F,H,I). Therefore, we believe

that C-terminal truncation of REEP5 did not cause destabilization or misfolding of the protein, although the effects could be due to the deletion of the conserved APH domain. To continue, we understand that disulfide bonds are less likely to be formed in a reductive cytosolic environment. However, two cysteine residues at position 59 and 118 are located in the transmembrane domains, leaving a possibility that covalent disulfide bonds could contribute to the dimerization process of REEP5. Furthermore, a corresponding accumulation of REEP5 monomers in response to an increased concentration of DTT was observed in the expression levels of endogenous REEP5 (Fig. 5C, indicated by orange diamonds). However, the same effect was difficult to observe in the exogenous expression levels of REEP5-V5-6xHis (indicated by blue diamonds) due to saturated expression levels of exogenous REEP5-V5-6xHis.

8. The authors used the entire Fig. 5 to demonstrate that REEP5 interacts with other ER shaping proteins. Given that similar results have been shown when ATL was identified, it is not clear why these results are informative. One may argue that the interaction between REEP5 and Climp63 is unexpected. Even if the interaction is real, the relevance is not addressed.

Response: We appreciate these comments and understand the reviewers concern about how this data was presented and the limited context to why we feel they are so important to this study. The known interactions of these family members of ER shaping proteins led us to document the interactions of specific family members within cardiac muscle cells; and identified the major family members present in the cardiomyocyte and showed that they form interactions. These results are critical to our understanding of how these proteins function in the specialized SR of myocytes. As discussed in the Discussion, not only do these SR/ER shaping proteins interact with one another to regulate SR/ER morphology, they also cooperate with one another to ensure proper SR/ER function (Lines 605-619). In fact, in our study, we show that disruption of these interactions through REEP5 depletion resulted in changes in the physiological concentrations of other SR/ER shaping proteins in cardiac myocytes, suggesting significant roles and cooperation between REEP5, RTN4, ATL3, and Climp63. In addition, our results support previous experiments showing interactions among SR/ER shaping proteins in a mammalian cardiac context, further emphasizing the importance of these protein-protein interactions not only in the ER, but also the SR. Furthermore, as discussed in the Discussion (Lines 619-632), the relevance of the newly founded interaction between REEP5 and Climp63 is explained, although unexpected, but this interaction may be important for cardiac SR structure and biology as cardiac myocytes are fundamentally more complex than yeast and Xenopus models.

9. In the final figure, the authors try to argue that there is certain expression correlation between ER shaping proteins. However, in panel B, even the loading controls are quite different from each other, it would thus be difficult to conclude that depletion of REEP5 actually causes decreased levels of Rtn4 and Climp63. Furthermore, changes in panels A and B look quite different.

Response: We thank the reviewer for these comments. See also Reviewer 2, Point 2. In the revised paper, we have quantified all immunoblots, and standardized them to controls. The newly organized data is presented in the revised figures and text.

10. Finally, the correlation between REEP5-related ER stress and cardiac phenotypes has not been tested. It would strengthen the conclusion if ER stress to be monitored in zebrafish with REEP5 depletion.

Response: We appreciate these suggestions and absolutely agree with the reviewer that this is an excellent point. However, while this would be a highly exciting set of studies to pursue, due to the limitations in availability of anti-zebrafish antibodies, monitoring ER stress with REEP5 depletion is not feasible at the current time. However, we have discussed within the text that clarifying these roles in an in vivo knockout model would be the next logical steps (Lines 659-660).

Minor concerns:

1. Line 362, Climp63 is not found in yeast.

Response: We apologize for this statement and have removed the sentence in our revised manuscript.

2. Line 388-393, it is confusing here. Did the authors try to show that REEP5 and Rtn4 form oligomers or they interact with each other. Ref. #29 shows that upon cross-linking, Yop1p or Rtn1p forms SDS-resistant oligomerization.

Response: We appreciate the reviewer's comment and apologize for the confusion. We meant that REEP5 and Rtn4 physically interact with each other based on our co-immunoprecipitation experiments (Lines 443-453).

Reviewer #2 (Remarks to the Author):

Lee et al. have discovered a new and previously unappreciated role for REEP5 in the tubular endoplasmic reticulum, cardiomyocyte and heart development. They use a variety of techniques to define this role and interactions. While they do not specifically determine the mechanism involved (future studies) this study does provide an important advance to the field.

1. My main concern is the statistical methods used. 1-way ANOVAs were used for normally distributed parameters. This method is appropriate when determining if a single treatment produces an effect; however, when comparing two or more treatments (e.g., shREEP5 vs. sh-scrambled) a 2-way ANOVA is more appropriate.

Response: We agree with the reviewer's comment and we have revised the statistical assays. Specifically, we have performed our statistical analyses again with 2-way ANOVA and have reflected these changes in our Figure Legends and Methods sections (Lines 937-943).

2. Fig. 1I: The authors state, “REEP5 levels were the highest in mouse ventricle”, but do not provide quantification. They state, “the predominant form in the ventricle was the 17kDa monomer of REEP5” and 1 sentence later state “while the monomer and trimer was the prevailing form in the ventricle”. Please correct this inconsistency and quantify REEP5 monomer, dimer and trimer levels.

Response: We appreciate these comments and have now included quantification of the immunoblot analysis of REEP5 shown in Fig. 1G to demonstrate the levels of the different oligomeric forms of REEP5 in the various tissues (Supp. 1B). We have also corrected this inconsistency in our Results.

Minor:

Fig 1H Figure Legend: The phrase, “ubiquitously detected in many ... tissues” is contradictory. Please remove “ubiquitously”.

Response: We appreciate the reviewer’s comment and have now corrected this redundant sentence in the Figure Legend.

Reviewer #3 (Remarks to the Author):

1. In the introduction, the phylogeny of the REEP proteins is mentioned as background, but in the results section it is phrased as though this was determined in this study. This should be clarified.

Response: We understand the reviewer’s concern and we apologize for this confusion. We have now clarified our bioinformatics work and in our revised manuscript, we have made the appropriate changes and elaborated on the experimental details in the Results as well as the Figure Legend (Lines 163-168; 1329-1334).

2. There needs to be more detail about the membrane yeast-two hybrid so that a reader can assess what is in front of them. The text says Ost1-Nubl is the positive control, and yet interaction with the control is used as proof of a result? This is either poorly worded or something is incorrect. What is it you want me to see on the yeast plates that tells me the REEP5 N-terminus is cytoplasmic? What does 5-FOA tell you in this particular setup? A diagram would be helpful.

Response: We appreciate these comments and agree that a detailed description of the membrane yeast two-hybrid experiments was needed. In this revised manuscript, we have ensured to provide a more detailed description and interpretation of the membrane yeast-two hybrid experiments in our study within the Methods section (Lines 756-768) Figure legend (Lines 1160-1166), and Results (Lines 173-184).

3. There are numerous Western blots throughout the paper without any quantification. Density quantification should be used at a minimum to backup the qualitative statements

about protein regulation and levels.

Response: See Reviewer 2, Point 2.

4. In the Figure 1 legend – there is an asterisk to denote REEP5 expression in ubiquitous tissues. Did the authors mean to say REEP5 is ubiquitous and the only REEP in cardiac tissues denoted with an asterisk?

Response: We thank the reviewer for pointing out this issue. In Fig. 1E, the asterisk was placed to indicate that REEP5 is the dominant REEP in cardiac tissues in both fetal and adult heart. However, we can see how together with the next panel it would be confusing. As a result, we agree removing these asterisks will streamline the presentation for readers. In Fig. 1F, the asterisk was meant to denote the alleged number of REEP5 molecules in the oligomers detected, i.e. one asterisk indicates a monomeric form of REEP5, two asterisks indicate dimeric form of REEP5, and three asterisks indicates trimeric form of REEP5. In our revised manuscript, we have ensured this is clear within the Figure Legend 1.

5. Line 260 – the authors report that cell death is not causing the morphological changes in CMNCs where REEP5 is depleted, and yet on Line 279 the authors conclude ER-dependent cell death is occurring. How do the authors reconcile this?

Response: We agree with the reviewer that this is a relevant point and one we need to explain more clearly. We used the chemical inhibitor, z-vad-fmk, to inhibit caspase activity and to prevent possible vacuolization and membrane blebbing that occur with late-stage apoptotic cell death. This treatment was to demonstrate that the vacuolization we observe is not solely due to the cell dying, but rather specifically due to REEP5's role in SR/ER structure maintenance. However, we have also shown that upon REEP5 depletion, we are starting to activate ER-dependent cellular pathways including ER-dependent apoptosis, as demonstrated by our caspase-12 immunoblots (Fig. 3G). In conclusion, we have shown that REEP5 induced SR vacuolization is a specific phenotype due to the REEP5 depletion, however, if this phenotype persists, this leads to activation of ER-dependent cellular pathways including unfolded protein response as well as ER-dependent apoptosis. We have included sentences to discuss this potential pathway within the revised Discussion (Lines 291-297).

6. In the immunolocalization studies, given the ER is almost ubiquitous throughout the cell, it's hard to evaluate where there is true co-localization (in the same place through interaction) versus co-compartmentalization since the ER is everywhere and the signal is going to overlap with other structures in the cell. Especially in Figure2 – this looks like they are expressed in similar domains, but have a lot of non-overlap as well. How meaningful is this? What happens to the sarcomere structure in TEM upon REEP5 depletion?

Response: We appreciate these comments and understand the reviewer's concern. This is a valuable point by the reviewer. The SR/ER is the largest membrane bound organelle within the cell and thus will form contact sites with other organelles within the cell. However, Our 3D reconstruction and

imaging line scan analyses (Fig. 2C) both demonstrate co-localization between REEP5 and SERCA2, suggesting REEP5 localization to the SR in the myocyte. Both detailed analyses show that areas where SERCA2 signals are higher are complemented by increases in REEP5 signals, showing significant co-localization as calculated by Imaris (Pearson co-localization coefficient 0.59 ± 0.02 with an average of $53.46\% \pm 0.86\%$ area of co-localization between REEP5 and SERCA2, and an average of $63\% \pm 2.4\%$ of REEP5 co-localized with SERCA2 and an average of $75\% \pm 1.7\%$ of SERCA2 co-localized with REEP5). These detailed statistical analyses demonstrate that we are able to differentiate between co-localization versus co-compartmentalization using Imaris. In addition, as suggested within this reviewers comments, we did perform new TEM imaging of sham and knockdown of REEP5 in myocytes and the analysis of REEP5 depleted myocytes show deformed SR membranes, SR vacuolization and deformed cardiac t-tubules (Fig. 4C).

7. On this note, in Supplemental Figure 3 – what is different between “merge” and “overlay”? The overlay shows there is a lot of non-overlapping expression, in fact more non-overlap than overlap. In the end, I am convinced they are both expressed in the same cell, but not sure what the localization tells me beyond that.

Response: We understand the reviewer’s concern and apologize for the confusion. The overlay is the signal and region that Imaris detected as being significantly co-localized or overlapped between the two proteins. Whereas the merge 3D reconstructed model show all signal detected separately. In our study, we show “overlay” to demonstrate the area of co-localization only between proteins of interest and “merge” to demonstrate the complete 3D reconstruction model with individual channels detected and built by Imaris.

8. I was specifically asked to comment on the zebrafish work in this manuscript. The authors use one ATG translation blocking morpholino to REEP5 to make claims about the role of this protein in cardiac function and development. Based on the standard of the zebrafish community, one morpholino is not sufficient due to the concern of off-target effects. It is good to see that the protein is depleted on the Western, but no rescue experiments are performed to determine what is off-target and what is not.

Response: We absolutely agree and understand the reviewer’s concern. Off-target effects are always a concern in morpholino-based knockdown experiments. However, in our revised manuscript, we have followed the journal guidelines regarding the use of morpholino use in zebrafish as recently published (Stainier DYR et al. 2017), as suggested by the Senior Editor. In order to assess off-target effects, we also investigated the effect of the REEP6 morpholino sequence. As detailed in our manuscript, REEP6 is the most similar protein to REEP5 in zebrafish, sharing 92% homology. In fact, functional analyses of REEP6 morphant embryos revealed no observable cardiac phenotypes and appeared otherwise normal and healthy, further supporting the specificity of our original REEP5 experiments (Figure 7 & Supp. 7). In addition, we have also increased our total sample size number of morpholino embryos and scrambled controls analysed to ensure consistent observations in this revised manuscript. We do feel comfortable that there is sufficient evidence of the specific role and phenotypes associated with

REEP5 depletion in the zebrafish heart. Nonetheless, we include several sentences in the Discussion as to caveats to any morphant results and that ongoing experiments will continue to refine the specific phenotypes of REEP5 deficiency in differentiated mammalian cells (Lines 656-660).

9. Morpholinos are notorious for causing developmental delays and off-target effects. Couple that with the fact that the cardiac system is particularly easy to perturb if development, in general, is affected, I find it hard to believe any of the cardiac function data as it stands. Given the ease of CRISPR-Cas9 technology, a knockout of REEP5 would be most helpful, and a targeted knockout in the heart would be ideal for these experiments to be meaningful.

Response: We agree with the reviewer that a CRISPR-Cas9 mutant fish would be a valuable tool; as would a conditional, tissue-specific knockout mouse. Both of these valuable models would provide immeasurable specificity and quality control to this field of enquiry. However, we feel that they are beyond the scope of this manuscript.

10. Other issues with the zebrafish work – it is not possible to determine the looping status of the hearts comparing D to E or F to G. In K-N hearts are seen as delayed in the cone stage, or looped, but in J the authors claim hearts remain linear. Are they saying the hearts loops and then become linear over time? Or that some heart fail to loop and remain midline?

Response: We agree with the reviewer that it is hard to determine the looping process of the heart based on the images presented. In our study, we have noticed that REEP5 depleted hearts failed to loop in vivo accompanied by atrial and ventricular swelling compared to the control embryos, however, this can be a difficult phenotype to quantify given the above-noted concerns. Accordingly, in this revised manuscript, we agree with the reviewer and decided to remove these images.

11. The immunofluorescence in the zebrafish heart is not informative. Higher resolution imaging would be required to say anything, and given the ubiquitous nature of the ER, it's not clear what can be obtained from this type of analysis as presented.

Response: Here, we are also in agreement with the reviewer. Similar to above, in this revised manuscript we have removed these immunofluorescence images. We now also provide additional transmission electron microscopy experiments on REEP5 MO morphants to provide in vivo ultrastructural analysis of REEP5 deficient myocytes in the zebrafish heart (Fig. 7D,E,F,G). Taken together, these additional experiments demonstrate the critical role of REEP5 in regulating SR integrity, normal cardiac development and function, and its impact in the progression and development of cardiac disease.

END OF COMMENTS

=====

Reviewers' Comments:

Reviewer #1:

Remarks to the Author:

The authors have addressed most of my concerns. The manuscript is now appropriate for publication in Nature Communications. I just have one more comment. The interaction between REEP5 and Climp-63 was discussed as for stabilizing edge curvature of sheets. However, it has been reported that Climp-63 plays an SR-specialized role (PMID: 27562070).

Reviewer #4:

Remarks to the Author:

I have looked at the comments by reviewer#3 and the response by the authors and I have read that part of the manuscript that deals with the zebrafish experiments (so I have not looked at the rest of the manuscript in great detail).

It is my opinion that the zebrafish experiments as described in the revised version are NOT suitable for publications. The main reason is that they do not follow the guidelines as described in the PLOS genetics paper by Didier Stainier et al. This was already correctly indicated by reviewer#3 and these concerns have not been addressed adequately in the revised version.

The guidelines clearly state that a morpholino knock down phenotype should be compared to the phenotype of a genetic loss-of-function mutant of the same gene. This comparison, which is an important step since it is known now that morpholinos can create off-target effects, was not done in this manuscript.

In addition, the authors have only used one morpholino and I did not see any rescue experiments (also included in the guidelines).

Something else that worries me is that the reep5 knock-down in zebrafish gives a very general developmental defect affecting many tissues. The heart is known to be very sensitive to such general developmental defects. This makes the follow up work less relevant since the heart related defects could be a consequence of the general developmental defect and not due to a specific role of reep5 in the heart.

Reviewers' comments:

Reviewer #1 (Remarks to the Author):

The authors have addressed most of my concerns. The manuscript is now appropriate for publication in Nature Communications. I just have one more comment. The interaction between REEP5 and Climp-63 was discussed as for stabilizing edge curvature of sheets. However, it has been reported that Climp-63 plays an SR-specialized role (PMID: 27562070).

Response: We appreciate these comments and have now included this important reference in our revised manuscript (Lines 665-668).

--

Reviewer #4 (Remarks to the Author):

The guidelines clearly state that a morpholino knock down phenotype should be compared to the phenotype of a genetic loss-of-function mutant of the same gene. This comparison, which is an important step since it is known now that morpholinos can create off-target effects, was not done in this manuscript.

Response: We appreciate these comments and understand the reviewer's concerns. In our revised manuscript, we have made sure to follow the journal guidelines regarding the use of morpholino use in zebrafish (Stainier DYR et al. 2017). Specifically, we have generated CRISPR/Cas9-mediated REEP5 loss-of-function crispants (Shah et al. 2015 Nature Methods) and performed dose response gRNA condition experiments (1x, 2x, and 3x gRNA), together with time course data in the developing embryos, matched with general histology and cardiovascular functional assessments. Importantly for this study, the results phenocopy the earlier cardiac dysfunction we observed in the morphants (New Figure 7 & New Supplementary Figure 8).

Something else that worries me is that the reep5 knock-down in zebrafish gives a very general developmental defect affecting many tissues. The heart is known to be very sensitive to such general developmental defects. This makes the follow up work less relevant since the heart related defects could be a consequence of the general developmental defect and not due to a specific role of reep5 in the heart.

Response: We agree with the reviewer that off-target effects are always a concern in morpholino-based knockdown experiments. In our revised manuscript, we have generated CRISPR/Cas9 mutant fish to prove the specific roles of REEP5 in the heart. In addition, our findings in zebrafish are entirely consistent with our mouse studies. Taken together, these additional experiments demonstrate the critical role of REEP5 in regulating SR integrity, normal cardiac development and function, and its impact in the progression and development of cardiac disease. Nonetheless, we have ensured that clear limitations to these studies is mentioned in the text (Lines 695-700).

END OF COMMENTS

=====

Reviewers' Comments:

Reviewer #4:

Remarks to the Author:

In this revised version of the manuscript 'REEP5 depletion causes sarco(endo)plasmic reticulum vacuolization and cardiac functional defects', the authors have included additional experiments to address the in vivo role of REEP5 in cardiac development and function. My previous concerns were related to the experiments in which antisense morpholinos were used in zebrafish embryos to address the role of reep5. These knock-down experiments did not comply with the guidelines for gene function analysis published in PLOS genetics by Stainier et al, which are widely accepted in the field. Since the new experiments are still not according to these guidelines, it is still my opinion that the zebrafish experiments as described in the revised version are NOT suitable for publication.

Specific comments:

1. My previous comment was that the guidelines clearly state that a morpholino knock down phenotype should be compared to the phenotype of a genetic loss-of-function mutant of the same gene. In this revised version the authors did not generate a genetic zebrafish mutant line for reep5 to analyse its phenotype. Instead the authors performed sgRNA/Cas9 injections and analysed transient phenotypes in the F0 injected embryos. These so-called crispants are transient knock-down experiments and come with the same limitations as antisense morpholino injections and therefore cannot replace a stable mutant line as indicated in the guidelines. Because the lack of a stable genetic mutant line to confirm the phenotypes observed in transient knock-down experiments, the revised manuscript does not comply to the published guidelines.
2. In my previous comments it was indicated that the authors have only used one morpholino targeting reep5 and I did not see any rescue experiments (also included in the guidelines). Although being of lesser importance in case a genetic mutant is used, these concerns were not addressed by the authors.

Specific point-by-point response:

=====

Reviewers' comments:

Reviewer #4 (Remarks to the Author):

1. My previous comment was that the guidelines clearly state that a morpholino knock down phenotype should be compared to the phenotype of a genetic loss-of-function mutant of the same gene. In this revised version the authors did not generate a genetic zebrafish mutant line for reep5 to analyse its phenotype. Instead the authors performed sgRNA/Cas9 injections and analysed transient phenotypes in the F0 injected embryos. These so-called crispants are transient knock-down experiments and come with the same limitations as antisense morpholino injections and therefore cannot replace a stable mutant line as indicated in the guidelines. Because the lack of a stable genetic mutant line to confirm the phenotypes observed in transient knock-down experiments, the revised manuscript does not comply to the published guidelines.

Response: We understand the reviewer's concern. We argued previously that the guidelines did afford latitude with experimental details. However, in this revised version, we have generated a sequence-confirmed stable reep5 homologous CRISPR mutants in zebrafish (see Editorial Review Figure 1 below, panels a,b) and include this data in our submitted revised manuscript. The reep5 homologous mutants did not display major developmental defects or the complete lethality we had observed previously in the other two strains of fish. These differences in phenotype reflect the reviewers concern that morpholino-based studies need to be carefully interpreted. We assessed expression of REEP5 and its associated protein complexes in these fish hearts and showed that RTN4 in reep5 CRISPR mutant hearts were all elevated, suggesting potential compensatory mechanisms regulated by RTN4 in response to the loss of REEP5 in zebrafish (panel d). However, importantly, we were able to induce functional differences between the strains, where the reep5 germline mutants DID display acute differential responses to verapamil, a voltage-gated Ca²⁺ channel blocker (panel c). All of these data are now included and discussed in the revised manuscript (Revised manuscript figure 7 and pages 18-20).

In addition, and in parallel to our zebrafish studies, we also performed in vivo AAV9-mediated REEP5 depletion in mice and examined cardiac function (Editorial Review Figure 2, panel a). We showed that these mice presented with lethal dilated cardiomyopathy phenotypes including myocardial pathology (Editorial Review Figure 2, panel b) and increased cardiac fibrosis (Editorial Review Figure 2, panel c). In vivo echocardiographic cardiac function assessment further demonstrated significantly compromised systolic cardiac function with reduced ejection fraction and cardiac output (Editorial Review Figure 2, panels d,e). We also demonstrated increased expression levels of ER stress markers (GRp78, GRp94, XBP1, ATF4, CHOP) as well as increased expression levels of cleaved caspase12, indicating activation of ER-dependent cell death in vivo in REEP5 depleted mouse hearts (Editorial Review Figure 2, panel f). Altogether, we have made significant additions to address the in vivo role of REEP5 using a multitude of models. We include all of these studies in the revised manuscript (Revised manuscript figure 8 & pages 21-23).

Editorial Review Figure 2. (a) immunoblot analysis of AAV9-mediated REEP5 depletion in mouse hearts. (b) H&E and (c) Masson's trichrome histological analyses of AAV9-mediated REEP5 depleted myocardium. (d) *in vivo* echocardiographic cardiac function assessment of AAV9 REEP5 shRNA injected mouse hearts. (e) Echocardiographic M-mode and B-mode measurements. (f) Immunoblot analysis of cardiac ER stress markers in AAV9-mediated REEP5 depleted mouse hearts.

2. In my previous comments it was indicated that the authors have only used one morpholino targeting reep5 and I did not see any rescue experiments (also included in the guidelines). Although being of lesser importance in case a genetic mutant is used, these concerns were not addressed by the authors.

Response: We appreciate the reviewer's comment and understand the concern. As discussed above, we have now added *in vivo* rescue experiments by injection of wild type REEP5 mRNA in the morphant embryos. We showed that co-injection of REEP5 MO and REEP5 mRNA efficiently rescued the cardiac dysfunction phenotypes observed in REEP5 MO embryos, resulting in significantly improved heart morphology, reduced atrial-ventricular dyssynchrony, and restored heart rate with an apparent healthy ratio of 1:1 atrial:ventricular contractions between the chambers in the rescued morphants (Editorial Review Figure 3, panel a-c). These data further highlight the specificity of our morphant experiments, and are now included in our revised manuscript (Revised manuscript supplementary figure 6n-r and page 18).

Editorial Review Figure 3 (a) REEP5 mRNA rescue experiments by co-injection of REEP5 mRNA and REEP5 MO in zebrafish embryos. (b) Video imaging analysis shows corrected cardiac beating rhythms with normal atrioventricular contractions in the rescued embryos. (c) Heart rate and optical video imaging analyses shows restored heart rate and reduced cardiac beating dyssynchrony in the rescued embryos.

In addition, we have included a 5-base mismatch MO study (which is also included in the guidelines) by designing and injecting the REEP6 MO sequence (which has 92% homology to REEP5) in the embryos. The REEP6 morphant embryos showed no observable cardiac phenotype and appeared healthy with normal, consistent heart rate throughout development (Editorial Review Figure 4, panel a). We have included this data in our revised manuscript (Revised manuscript supplementary figure 6d and page 17)

Reviewer #3 (Remarks to the Author):

1. Using a second morpholino AND rescue experiments to provide confirmation in line with Stainier et al. or analyzing stable mutants for REEP5 that they should be on their way to having at this point.”

Response: As detailed above, we have added significant revisions to the fish models, including the sequence-confirmed stable reep5 homologous CRISPR mutants in zebrafish in our revised manuscript. In addition, we have also included in vivo rescue experiments by injection of wild type REEP5 mRNA in the morphant embryos as well as a 5-base mismatch MO study in our revised manuscript. These data are included in our Revised Manuscript Figures 7,8 & pages 17-23.

END OF COMMENTS

=====

Reviewers' Comments:

Reviewer #4:

Remarks to the Author:

In this revised version of the manuscript the authors have performed additional *in vivo* loss-of-function experiments in zebrafish and mice. For the zebrafish, the authors generated a stable line with a *Reep5* mutation and analysed the homozygous mutant embryos. This new information is very important to understand the role of *Reep5*. The authors describe that the zygotic homozygous mutant embryos do not display any embryonic phenotype and therefore do not recapitulate the morphant or crisprant phenotypes. While this is important data to include in the manuscript, it is surprising that this discrepancy between the knock-down and the knock-out phenotypes are mostly ignored in the discussion, which leaves the reader in confusion. What is in the opinion of the authors the explanation for the observed discrepancies? There is no description of the consequence of the germ-line mutation on the *Reep5* protein (except the absence of a band on the western blot). The authors mention briefly the recently described compensation mechanisms but they have not tested if this could explain the different phenotypes by injecting the MO into the homozygous mutant embryos (should not give a phenotype if genetic compensation is responsible for the absence of phenotypes in the mutant). Likewise, the authors do not discuss the possibility that maternal mRNA compensates for the loss of zygotic *reep5* mRNA. The authors observed a convincing rescue with the *reep5* mRNA (which has been modified so that it lacks the MO binding site) suggesting that the MO-induced phenotype is indeed caused by loss of *reep5*. Have the authors performed some more detailed analysis (besides heart rate) on cardiac function or structure in the homozygous mutant fish. What about SR structure in the homozygous mutant fish? The readability of the manuscript would greatly benefit from a more elaborate discussion on this topic.

Likewise, the authors present new data using adeno virus mediated knock-down of *reep5* in mice that show a drastic effect on cardiac function. However, a previously described rat knock-out model, which the authors refer to, displayed a much weaker and non-lethal cardiac phenotype. Again, this discrepancy is somewhat ignored in their discussion.

Specific point-by-point response:

=====

Reviewers' comments:

Reviewer #4 (Remarks to the Author):

1. In this revised version of the manuscript the authors have performed addition in vivo loss-of-function experiments in zebrafish and mice. For the zebrafish, the authors generated a stable line with a Reep5 mutation and analysed the homozygous mutant embryos. This new information is very important to understand the role of Reep5. The authors describe that the zygotic homozygous mutant embryos do not display any embryonic phenotype and therefore do not recapitulate the morphant or crispant phenotypes. While this is important data to include in the manuscript, It is surprising that this discrepancy between the knock-down and the knock-out phenotypes are mostly ignored in the discussion, which leaves the reader in confusion. What is in the opinion of the authors the explanation for the observed discrepancies? There is no description of the consequence of the germ-line mutation on the Reep5 protein (except the absence of a band on the western blot). The authors mention briefly the recent described compensation mechanisms but they have not tested if this could explain the different phenotypes by injecting the MO into the homozygous mutant embryos (should not give a phenotype if genetic compensation is responsible for the absence of phenotypes in the mutant). Likewise, the authors do not discuss the possibility that maternal mRNA compensates for the loss of zygotic reep5 mRNA. The authors observed a convincing rescue with the reep5 mRNA (which has been modified so that it lacks the MO binding site) suggesting that the MO-induced phenotype is indeed caused by loss of reep5. Have the authors performed some more detailed analysis (besides heart rate) on cardiac function or structure in the homozygous mutant fish. What about SR structure in the homozygous mutant fish? The readability of the manuscript would greatly benefit from a more elaborate discussion on this topic.

Response:

i) In our revised manuscript, we have now also included optical imaging and movie analysis to examine heart function in the reep5 homozygous mutant fish compared to REEP5 morphant zebrafish hearts (Editorial Review Figure 1, panels e-g & Supplementary Movies 8,9; and included in revised manuscript Fig. 8d,e,i,j). All of these comments and figures are now included in a revised version.

ii) We appreciate the reviewer's comment and agree with the reviewer that it is important to discuss the discrepancies between our knockdown and knockout zebrafish models. The observed phenotype discrepancies between our morphant, crispant, and germline homozygous reep5 mutant models in zebrafish can likely be attributed to the difference between knockdown and knockout experimental approaches. Morpholino and crispant-driven knockdown experiments in zebrafish embryos allow investigations of acute embryonic phenotypes in fish embryos, however, does not take into account the possibility of long-term systemic effects of such interventions including possible compensatory mechanisms. On the other hand, the generation of

genetically edited homozygous mutants provides a critical window for physiological adaptation and possible functional compensation through breeding generations. For our analyses, we examined F4 generation fish. In fact, our immunoblot analysis of the closest member to REEP5 with functional similarity, RTN4, shows a significant upregulation of RTN4 in the germline homozygous reep5 mutants, however, the same phenomenon is not observed in REEP5 morphants (see Editorial Review Figure 1 below, panel a; and included in revised manuscript Fig. 8l and Supplementary Fig. 6q). These findings indicate that the consequence of germline reep5 deletion is, perhaps, masked by the upregulation of RTN4, as well as other related proteins, in the germline reep5 zebrafish mutant hearts. All of these comments and figures are now included in a revised version.

iii) Furthermore, as requested, we have now experimentally tested the idea of genetic compensation by injecting REEP5 MO into our homozygous reep5 CRISPR mutants and show no observable cardiac phenotype was detected in our germline reep5 mutants compared to apparent cardiac abnormalities observed in the REEP5 morphants (Editorial Review Figure 1, panels b-d; and included in revised manuscript Fig. 8g-k). These data further support the idea that genetic compensation is perhaps responsible for the absence of phenotype in our homozygous reep5 CRISPR mutants; an effect reported in the fish literature (Rossi et al. Nature 2015). All of these comments and figures are now included in a revised version.

iv) Lastly, to rule out maternally-contributed mRNA expression of REEP5 in the process of generating germline CRISPR reep5 mutants, we generated maternal zygotic mutants of reep5 by performing an additional cross with an identified homozygous female reep5 mutant to produce a maternal zygotic homozygote. These maternal zygotic homozygous mutants were used in our studies. We have now included this important detail in our Methods, Results and Discussion and expanded our discussion on all of the abovementioned topics (pages 19-21;25-26;38). Again, all of these comments and figures are now included in a revised version.

Editorial Review Figure 1. (a) Immunoblot analysis of RTN4 in REEP5 morphant and *reep5* CRISPR mutant hearts. (b,c) REEP5 morphant and *reep5* homozygous CRISPR mutant injected with REEP5 MO. (d) Phenotypic penetrance represented as percentage. (e-g) Optical imaging of REEP5 morphant, *reep5* CRISPR mutant, and REEP5MO+*reep5* CRISPR mutant zebrafish hearts.

2. Likewise, the authors present new data using adeno virus mediated knock-down of *reep5* in mice that show a drastic effect on cardiac function. However, a previously described rat knock-out model, which the authors refer to, displayed a much weaker and non-lethal cardiac phenotype. Again, this discrepancy is somewhat ignored in their discussion.

Response: *In our revised manuscript, as highlighted in comments above, we have now expanded our Discussion to address the phenotype discrepancy across the multiple models of knockdown and knockout (revised manuscript pages 25-27). In particular, we highlight that the in vivo AAV9 mouse work is entirely consistent with the mouse neonatal and adult cardiac myocyte experiments, the in vivo REEP5 zebrafish morphants, and REEP5 zebrafish crispants; all of which are acute knockdown experiments. Importantly, as we detailed above in earlier reviewer comment discussing knockdown vs knockout, the mouse experiments represented this acute knockdown vs a genetic deletion. As well, the discrepancy in phenotype severity between our mouse and the rat may be a species dependent phenomenon, as rat models of heart diseases (myocardial infarction and pressure overload-induced heart failure) in general have shown different contractility, and a slower disease progression compared to the same procedures in mice (Patten et al. Circ Heart Fail 2009; Milani-Nejad et al. Pharmacol Ther 2015; Camacho et al. Am J Cardiovasc Dis 2016).*

Nonetheless, we remain convinced that future studies identifying human mutation of reep5 in heart disease patients would provide immeasurable specificity and invaluable control to this field; until then, ongoing experiments will continue to refine the specific phenotypes of REEP5 depletion in the heart. Nevertheless, we have elaborated on this topic in our Discussion (revised manuscript page 27).

END OF COMMENTS

=====

Reviewers' Comments:

Reviewer #4:

Remarks to the Author:

The authors have included extra information that helps to explain the discrepancies between the morpholine knock-down and the genetic mutants in the zebrafish and extended the discussion. Most importantly, the MO injection in the mutant fish did not result in a cardiac phenotype and this suggests that a genetic compensation mechanisms is activated in the mutants. The new data and the textual changes address all my previous concerns